# Bridging the gap in customised housing design: Integrating a graphic user interface for user collaboration

**Micaela Raposo** ***, Sara Eloy¤, Miguel Sales Dias**

ISTAR, Instituto Universitário de Lisboa (ISCTE-IUL), Lisbon, Portugal

¤ Current address: Department of Interior Architecture, University of Antwerp, Antwerp, Belgium
* mmros@iscte-iul.pt

## Abstract

This research addresses housing customisation by using digital tools in co-design processes. This paper introduces the development of a Graphical User Interface (GUI) that assists the end-users' interaction with the design process. Although the participation of end users in the design process is considered essential, a communication gap persists between designers and end users, and there is a lack of tools to help inhabitants express their needs and desires. To bridge this gap, this research proposes using digital technologies to enhance end-users participation in the design process of their houses. In this paper, we show the results of the development and evaluation of an interface designed to help inhabitants to co-create their houses. We developed and tested a GUI for a housing co-design system. Interviews with professionals, housing cooperatives, and inhabitants informed the design process, allowing us to define user requirements and design tasks. The interface was tested with low and high-fidelity prototypes, receiving positive feedback from both experts and potential users. Architects were also involved in using the interface to comment on its usefulness for housing co-design. The tool demonstrated the potential to improve end-users' participation, contributing significantly to participatory processes in collective housing. This research ensures the tool's effectiveness by directly incorporating user input, aligning the interface with the user's needs and preferences.

## Introduction

End-user satisfaction with the built environment depends largely on how the space suits their needs. This is especially important in housing design, where the inhabitants should have a say in the design process. However, it is difficult for people who lack the technical knowledge to communicate their aims effectively, leading to a communication gap between designers and end users.

To address this issue, we propose using digital technologies to improve participatory processes to achieve customisation in housing design through co-design. Such digital technologies should incorporate a realistic representation of the design solution to empower the

**Data Availability Statement:** All Supporting Information files are available from the Zenodo database (accession number(s) 10.5281/zenodo. 12187613).

**Funding:** This work was supported by National Funds through Fundação para a Ciência e a

Tecnologia, I.P. (FCT), through the grant SFRH/BD/146044/2019, and under the ISTAR projects: UIDB/04466/2020 and UIDP/04466/2020. The funders had no role in study design, data collection and analysis, decision to publish, or preparation of the manuscript.

**Competing interests:** The authors have declared that no competing interests exist.

understanding of space by non-specialists and deliver design solutions that fit the aims of the inhabitants. Several studies have shown that the use of realistic representation of architecture, such as image rendering of 3D models, improves the understanding of space by non-technical [1–3]. Additionally, using generative design systems [4], enables the creation of a wide range of solutions that may respond to the diversity of end users and comply with construction regulations.

There are a few computational systems for designing customised housing [5]. Nevertheless, a large number are not accessible to a non-technical public and do not have user-friendly interfaces. As far as we could find in the literature, there is no evidence that end-users, i.e., inhabitants, were part of the development process of the mentioned systems. Therefore, there is a need to develop easy-to-use interfaces that respond to the needs of its users.

This research addresses the end-users interaction with the design process via a Graphical User Interface (GUI). In this paper, we raise the question "Can the inhabitants easily interact with the architectural design of houses in an informed way, to reach customised housing?". We hypothesise that a housing co-design digital tool can assist inhabitants in this task, as digital technologies improve the end-user understanding and have the potential to enhance their collaboration in the design process. A natural interface would allow inhabitants to easily interact with the design of their houses and, thus, achieve customised housing. Therefore, our goal is to define, prototype and test a graphical user interface for a housing co-design system that focuses on the user experience. We also aim to identify the impact of the inhabitants using a digital design tool in the scope of a co-design process and definition of customised housing.

Our work contributes to knowledge development because the co-design tool is defined based on requirements extracted from direct contact with potential users. In this way, we ensure that the interface meets the user's needs. By adopting a user-centred design approach, we aim to bridge the gap between designers and end users to allow the creation of truly customised houses.

In the next section, *Framework*, we present a framework on the participation and co-design concepts and the use of digital technologies for housing customisation. Section *Methodology* explains the methodology used for the definition and assessment of our graphical user interface. In the section *Collection of users needs by interviews* we present a brief description of the main insights from the interviews. In the section *Definition of the system* we describe the definition of the system based on the requirements extracted from the interviews. Next, in section *Prototyping and evaluation*, we present the results of the two phases of the prototype development and assessment. In *Discussion*, we discuss the results and, finally, in *Conclusion* the conclusion and paths for future work are presented.

## Framework

Customisation is becoming increasingly important as individuals seek products and services that meet their unique needs and preferences. This is especially true when it comes to designing houses. Housing is an essential good, and its customisation greatly impacts people's quality of life [6]. For houses to be truly customised, it's crucial that inhabitants participate in the design process.

Mihaela Zamfir highlights interdisciplinarity as a particular characteristic of community architecture [7]. This interdisciplinarity involves different fields, e.g. education, public services facilities, public space, and housing. This practice is also vast in scale, as it covers small and large-scale projects. Community participation has its roots in the work of architects such as Yona Friedman and John Habraken, pioneers in using flexibility and adaptability in architecture to empower citizens to have a role in decisions about the design of their living spaces.

Friedman's Architecture Mobile Manifesto [8, 9] embraced the idea of a flexible architecture to respond to the needs of its future inhabitants. This idea was represented in the *Ville Spatiale*, a conceptual utopian city developed featuring modular, reconfigurable structures on top of existing cities. Habraken's concept of "Supports" [10] addresses an alternative for mass-housing customisation. The building is decomposed into two entities: the support and the infill. The support is the building's envelope, which represents the framework for providing dwellings that can be defined and altered independently from others. The architect presents and designs the support's constraints. The infill is the part within the principles defined by the architect which is chosen by the inhabitants.

Participatory design allows for a customised and user-centric approach to housing design. It ensures that the dwellings meet the users' needs and preferences, leading to more satisfied homeowners and a better quality of living. Such an approach to architectural design concerns the active involvement of end-users in the design process, allowing them to contribute with their ideas and feedback to create solutions that meet their needs. Participatory design has several benefits, including increasing user satisfaction, a sense of belonging, and community empowerment [11, 12].

As Sherry Arnstein defines, participation is a political process of power redistribution, and different participation levels can be applied. According to the author, participation goes from the mere information and consultation of the user's needs to a more active role of the community in the decision-making process [13]. Examples of different cases (and levels) of end-users involvement in the design process since the 60s are the work of the architects Ralph Erskine, Álvaro Siza Vieira and John Turner.

Ralph Erskine's work was considered central to the community architecture movement [14]. One of his best-known works is the Biker Wall, a housing estate constructed in Newcastle between 1969 and 1975. The community was involved in the development of the project through consultation sessions to provide the architects' team with the experience of what it is to live in a community. In such sessions, citizens could express their aspirations and feelings so that the architects could interpret and translate them into the design proposal [15]. Another example of community involvement in housing design projects is the work of Álvaro Siza Vieira with the *Punt en Komma* housing complex developed in the Dutch city of Hague between 1984 and 1988. The future inhabitants were invited to discuss the design by experimenting with the space themselves, using human-scale models. From the experience, the inhabitants produced a list of requirements for the architects to include in the design [16]. John Turner, on the other hand, highlights the advantages of the "Progressive Development" of popular settlements over official processes based on his experience with self-construction processes in Latin America, such as the case of *Pampa de Cuevas* in the outskirts of Lima, Peru [17]. The author believes that housing created by the community, which organises itself in voluntary associations, responds better to the needs of the inhabitants and the local culture.

However, although the three examples mentioned have different levels of community involvement, this power transfer is criticised and raises questions about what is an acceptable way of producing architecture. As a result, participatory processes can be neglected, raising doubts about their legitimacy [18]. Indeed, Erskine's Byker Wall was criticised for advocating being a participatory process, while the architects' ideas were masqueraded by the community involvement in consultation sessions [18]. Jeremy Till [19] states that in many cases there is a false participation, and that architects use technical language as a shield to present their ideas without being able to be actually discussed due to the lack of understanding of the community (non- experts in architecture). The author defends that one of the most important issues in achieving genuine participation is the formulation of a shared language between users and designers. Till highlights the need for new ways of communication, in which the architect

must be the "negotiator of hope" and act as much more than a technical facilitator [19]. Accordingly, other authors, such as Zamfir [7] and Habraken [20] state that a shift in the architect's role is necessary, as the architect should give increasing importance to social values.

Participatory design has been particularly successful in affordable housing projects, where community involvement has been shown to lead to better design outcomes and greater user satisfaction across the world in countries such as Colombia [21], Turkey [22], and Palestine [23]. However, as Davidson *et al.* [21] found, some participation levels are ineffective for user satisfaction for the lack of active involvement of the inhabitants in the design definition [13].

Community architecture is a continuing movement that is increasing to higher levels of involvement. Co-design represents a step further in participatory processes as it is an approach where the stakeholders collaborate in the decision-making process with the same authority level and contribute actively to the design proposal [24]. Brandt, Binder, and Sanders state that "making activities" help end-users in architecture to express their ideas [25]. However, the lack of technical knowledge and tools that help them give form to their wishes hampers the communication between designers and end-users [12]. Indeed, the lack of knowledge is assumed to be one of the main reasons architects do not involve the end-users in the design definition [26]. Architects also find participatory processes time-consuming [26, 27]. Authors such as Khalili-Araghi and Kolarevic [28] and Kwiecinsky and Słyk [29] recognise the fact that traditional processes with manual participation and co-design techniques are time-consuming, both for their analogic nature and for validating the designs produced by laypeople. Recognising the importance of end-user involvement, the authors highlight the need to address other ways of validating multiple design solutions. Digital technologies significantly impact participatory processes in architecture, as they can transform how people engage with the design. Immersive 3D visualisation (e.g., walkthrough 3D models) and the ability to show the consequences of user's actions in real-time are examples of how digital technologies improve the understanding of space and design possibilities by non-specialists in architecture [1–3].

Digital technologies also play a significant role in enabling customisation. Generative design is a computational process in which the computer generates multiple design solutions based on parameters. This process allows for the exploration of multiple design options in a short amount of time [4]. To accommodate the user's specific needs and generate design solutions that fit them, the user can adjust the parameters. Kwiecinsky and Słyk [29] state that generative design systems are necessary to validate design solutions created by the end-users since such systems are based on pre-validated design rules. In this way, it is possible to expand the viable solutions range and overcome the time barriers common in traditional participatory processes.

Different generative tools are available for architectural design based on cutting-edge technologies, e.g. machine learning, genetic algorithms and artificial intelligence. Examples of such design tools are TestFit [30] and Archistar [31], AI-based systems designed to create different solutions for buildings that comply with construction regulations and assess their feasibility in a detailed plot analysis [32]. With such systems, it is possible to customise designs, e.g., the distribution of living units. However, these tools are targeted at designers and other stakeholders such as builders and investors, and their interfaces are complex [32]. A generative system targeted for end-users in a co-design process needs user-friendly interfaces that do not require technical knowledge from the user.

The computerised version of the Segal method is an example of using a generative system to create customised housing that does not require technical knowledge from the end-user. The Segal method was created by the architect Walter Segal in the 1960s to allow self-builders to design their houses, becoming not only self-builders but also self-designers [33]. It consisted of a modular timber-framed set of panels that could be combined in almost any way to create

customised houses in a simple way for laypeople [34]. In the 1980s, John Frazer extended Segal's method to a computerised system, where a kit of parts could be assembled into a model on an electronic panel that could automatically scan the parts and generate drawings and three-dimensional views [35]. The system could also make calculations and simulations to give feedback to the users' input, made through the tangible interface, acting as a design assistant [34]. Frazer's computational tool embeds the architect's knowledge and experience through the design rules, thus ensuring the feasibility of the inhabitants' designs.

Other examples of rule-based generative design systems are the ones based, e.g., on shape grammars and parametric design tools. The grammar of Siza's houses at Malagueira [36] and Haiti grammar [37] are examples of shape grammars created to generate customised houses in a specific architectural style by introducing the user requirements. A similar approach to Frazer's generative system using a tangible interface is the HOPLA-Home Planner, developed by Kwiecinsky and Słyk [29]. The authors developed a generative system based on parametric design that responds to user actions using a tangible interface to move parts of physical models. However, the authors experienced difficulties with the tangible interface and a new version was created to be used on a touchscreen device.

Nonetheless, while digital tools have brought significant benefits to customisation and participatory and co-design processes, it is essential to address issues related to digital literacy. Developing user-friendly interfaces is crucial so inhabitants can easily interact with the design.

Some research prototypes based on generative design systems were developed to design solutions that respond to the user inputs introduced in the first steps of the system's use. Examples are MALAG [38], Barcode Housing System [39], i_Prefab Home [40], and ABC-based Customized Mass-Housing Generator [41]. These systems provide a set of parameters that can be adjusted to correspond to the user's needs. Another group of prototypes helps the user to create the house layout step by step according to the decisions taken through the design process. Examples of such generative design tools are HouseMaker [42], Group Forming [43], and ModRule [44].

Also, implemented commercial solutions such as IKEA Home and Kitchen Planner [45], Sweet Home 3D [46], and Room Sketcher [47] were developed. Such systems allow end-users to customise their own living spaces (including the layout, finishes, and furniture), designing them from scratch according to their preferences. Some of these tools do not incorporate construction regulations and can mislead users about the functionality and the construction viability of their designs.

Although effective customisation and participation need tools to combine the potential of generative design with user-friendly interfaces, the authors did not find evidence in the literature of the development of the tools mentioned involving potential users.

User-centred design aims to develop products that are easy to use. Actively involving end users in the development process and evaluating them iteratively leads to products that meet their users' real requirements and facilitate their satisfaction. Gonçalves, Fonseca and Campos [48] state that, although developing computational tools can be time-consuming, usability engineering offers benefits in terms of development cost, quality of the final product, and user satisfaction. Design problems can be detected at an early phase, which saves time and money. The authors explain that prototyping aims to reduce the time and cost of developing something that users can test.

Low-fidelity prototyping, e.g. paper prototypes, allows to quickly test features without spending too many resources in order to identify usability problems that can be solved at a preliminary phase [49]. On the other hand, high-fidelity functional prototypes allow to extract usability measures that paper prototypes do not allow, such as efficiency (which measures the time to perform a task), for allowing to simulate the dynamic feedback of the system [48, 50].

Figma, Adobe XD, and InVision are examples of tools that allow the creation of high-fidelity prototypes. What these tools have in common is their ability to create functional prototypes without any code and share them with other stakeholders. Figma [51] is a collaborative web app, although it also has a desktop application version. This codeless interface design tool allows the creation of high-fidelity prototypes with some interaction possibilities that simulate the feedback of the system whose interface is under development. Adobe XD [52] is another tool for User Interface (UI) and User Experience (UX) design. The clipboard-based workflow is similar to Figma and allows the design of different screens and connecting them, adding animations to create interactive prototypes. Similarly to both tools mentioned above, InVision [53] is also a web app that can create user interfaces for websites or apps for multiple devices, with screen connections that simulate the system's feedback to users' actions in a collaborative environment. Figma is one of the developers' most commonly used design tools for web interfaces and mobile applications for their ease of use [54]. The literature shows that it is used to develop interfaces in various fields, such as health and nutrition [55], digital tourism [56, 57], apps for ordering home services [58], and the financial sector for managing loans for agricultural activities [59]. In both of the mentioned examples, the authors used Figma as the prototyping tool within a user-centred design framework, using methods such as Design Thinking and lean UX in the iterative development process.

Using a user-centred design methodology is paramount to achieving an interface that meets users' needs. It is necessary to involve end users iteratively, using low and high-fidelity prototypes in various design development stages rather than just testing a final product.

## Methodology

For the definition of a graphical user interface that allows inhabitants to co-design their houses, we followed User-Centred Design principles [60] to accomplish the goal of conducting a research with a focus on the user experience. A ten-step iterative process was conducted, as depicted in Fig 1: (1) collection of users needs, (2) definition of user requirements, (3) definition of the system's features, (4) definition of the graphical user interface, (5) formative evaluation, (6) refine the interface, (7) heuristic evaluation, (8) second interface refinement, (9) summative evaluation, and (10) discussion with architects.

The first iteration involved collecting user needs (step 1), we interviewed professionals from architecture, urbanism, and social sciences to gather information on participatory processes and the potential of using digital tools. Furthermore, we interviewed housing cooperatives and inhabitants of these institutions (our potential users) to identify how participation occurred in this specific context and how a digital tool could have improved it. We focus on housing cooperatives as an example of a multi-family housing context, which would benefit from the proposed interface.

The results allowed us to extract the user requirements for the interface (step 2). Based on these requirements, the interface's features were defined (step 3), with which we could draw the graphical interface that users interact with (step 4). In the second iteration, we tested a low-fidelity paper prototype in a formative evaluation (step 5) with five potential users. According to their feedback regarding their satisfaction and the system's usability, we refined the interface and created a high-fidelity prototype to be tested again (step 6). In step 7, we conducted the third iteration by performing a heuristic evaluation with five interface design experts to identify and correct issues that potential users could possibly face. In step 8 we made a second interface refinement by correcting the issues identified by experts in the previous step. We proceeded to the fourth iteration, which includes steps 9 and 10. In step 9, a summative evaluation was performed with a more relevant number of potential users (inhabitants). Finally, in step

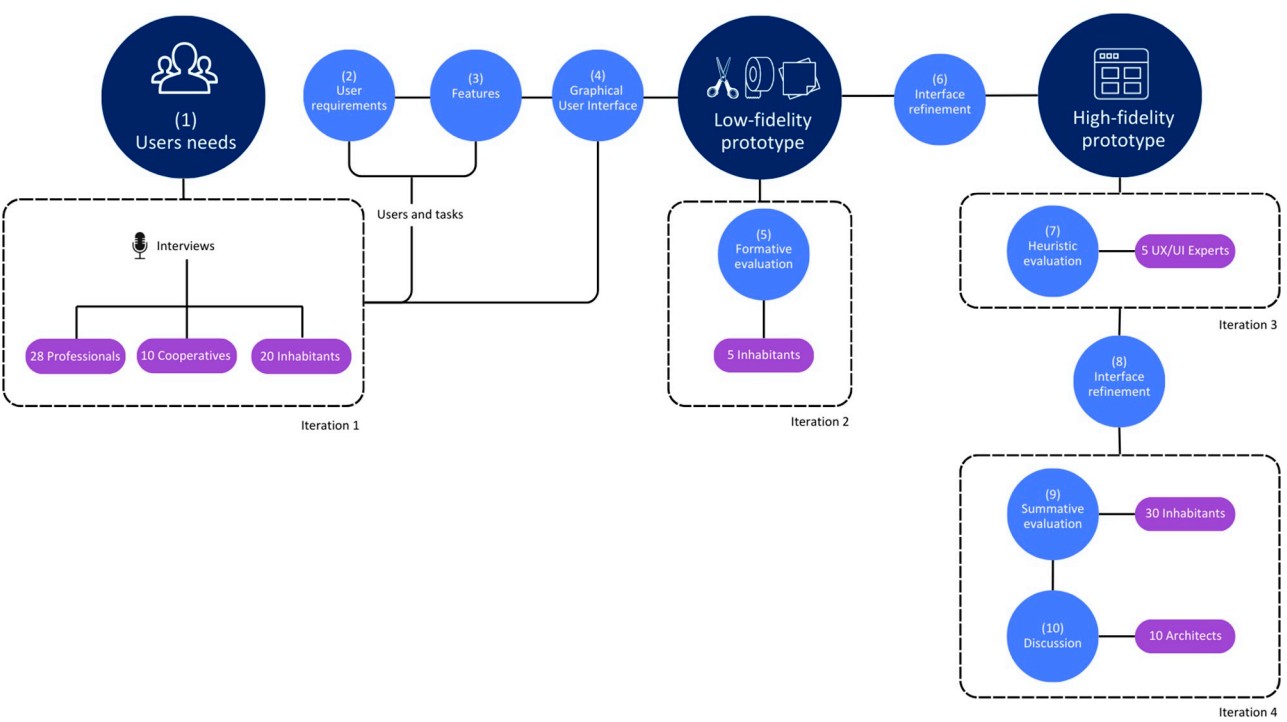

**Fig 1. Graphical user interface definition and testing process.**

10, a group of architects commented on the system after using the prototype to discuss, from their architectural expert point of view, the use of such a tool in a co-design process.

The method used for gathering the interviewees was snowball sampling. The professionals were selected according to their professional practice: architecture and urbanism, social sciences, and other professions related to housing or co-design. We identified in the literature and in an internet search professionals and architecture offices who were previously involved in participatory practices or who work within the mentioned areas. At the end of the interview, we also asked them to suggest other professionals who would fit the interview purposes, and we managed the contacts, keeping a balance between the number of national and foreign interviewees.

The institutions were selected through the search of Portuguese housing cooperatives. We contacted housing cooperatives and residents' associations across the portuguese territory, and the ten interviewed were the ones who agreed to collaborate on the research.

The inhabitants of housing cooperatives were recruited through a request to the cooperative representative to invite the inhabitants to participate in the interviews. The criteria was that the inhabitants should be the house's first ever tenants rather than someone who acquired the property later, through purchase from another tenant, as they did not participate in any phase of the design and construction process. Those interested in participating permitted the cooperative representative to provide their contact details so they could be contacted to schedule the interview.

All the participants agreed to participate in the study according to the user study protocol approved by Iscte—Instituto Universitário de Lisboa Ethics Committee. The professionals who participated in the interviews were recruited between June 1st 2021 and September 12th 2021. The representatives of housing cooperatives and inhabitants were recruited between

October 18[th] 2021 and November 29[th] 2021. The informed consent was obtained verbally and documented in video recording through the Zoom platform.

Regarding the interface evaluation (steps 5 to 10), the participants of the formative evaluation with the low fidelity prototype were selected through contact with the inhabitants of housing cooperatives that participated previously in the interviews. These inhabitants had previously expressed an interest in participating in the evaluation when they were asked during the interviews if they would like to be contacted again to see the result of the work developed after the interview in which they participated.

Regarding the evaluation with the high-fidelity prototype, the selected participants for the heuristic evaluation are recognised in the field as experts in user experience (UX) and User interface (UI). Five experts were selected in order to comply with the recommendations identified in the literature, which recommends the involvement of at least 3 to 5 experts [61]. Interface Design experts and architects were selected only for their professional practice, with age or gender not being considered in the choice of participants. The inhabitants who participated in the summative evaluation with the high-fidelity prototype were not the ones who participated in the previous evaluation and never had contact with the interface so as not to compromise the results. The recruitment was also made through snowball sampling, with the criteria of being between 25 and 45 years old, as this age range was identified during the interviews as the age of the potential users of the interface.

The participants of both evaluations agreed to participate in the experience and signed a written consent form. The participants of the formative evaluation (step 5) were recruited between October 13[th] 2022 and October 25[th] 2022. The participants of the heuristic evaluation (step 7), summative evaluation with inhabitants (step 8), and discussion with architects (step 9) were recruited between April 26[th] 2023 and May 31[st] 2023.

## Collection of user needs by interviews–step 1

### Interviews protocol

Fifty-eight interviews were conducted with three different groups: 28 professionals from architecture and urbanism, and social sciences areas; 10 housing cooperatives representatives; and 20 inhabitants living in houses from those cooperatives. Neither the age nor the gender of respondents was controlled. Men and women between 25 and 77 years old were interviewed. The general aim was to identify how participatory design processes occurred and could ideally occur and how digital tools can help these processes.

The Interviews with professionals focused on their opinions about three aspects: (1) the need for housing customisation, (2) the participation of end-users in the design process, and (3) the use of digital technologies in participatory processes.

The interviews with housing cooperatives aimed to identify how these institutions work regarding the design of new housing and whether inhabitants participated or expressed their willingness to participate in the design process of their houses.

From the interviews with inhabitants, we collected their perspectives on the process they were involved in regarding how they participated and how it could be improved. The questions were divided into three parts: 1) the participation process they were involved in, 2) their satisfaction with the process and the design outcome, and 3) how a digital tool could help them to participate more actively in the design definition. The third part aimed to define aspects of the tool, such as features (stages of the project in which they would like to participate), types of visualisation, and interaction that could facilitate a participatory task.

The interviews were semi-structured with open-ended questions, as the goal was to explore the issues addressed and not limit the answers to predefined options. However, in the third

part of the interviews with the inhabitants, closed-ended questions were made because the goal was to obtain specific data for defining the requirements of a digital tool. Therefore, along the questions, examples of types of visualisation, interaction, and devices to be chosen by respondents were shown.

The interviews were conducted online during 2021 and were recorded (audio and image—with the explicit consent of the interviewee) through the Zoom platform and lasted between 60 and 90 minutes.

The complete interviews protocol is provided as supporting information in S1 File. The results of part of the interviews are reported in [62] and other part is foreseen as forthcoming work. However, we provide the collected data in the S2 File and, in the following sub-sections, a description of the main insights taken from the interviews with each group is presented for context.

## Interviews with professionals from architecture and social sciences

The twenty-eight professionals interviewed are divided into two groups:

a. Eighteen professionals with experience in participatory processes. These interviews focused on the report of a specific case to understand how the process took place and what strategies were used.

b. Ten professionals who do not practice participatory processes. These interviews focused on the professionals' opinions regarding housing customisation, the inclusion of inhabitants in the design process, and how this could occur.

Fifteen Portuguese professionals were interviewed, and the remaining thirteen were from other countries. Respondents work with project typologies such as housing, public space, urban design, and public buildings.

Although all professionals agree that housing should be customised, the social sciences' professionals emphasise that housing is an essential good and that the possibility of customisation positively impacts the inhabitants' quality of life. However, some professionals say that customisation should not compromise the cost and flexibility of housing so that it can adapt to new demands.

Respondents recognise the lack of architects' openness as one of the main reasons that inhibit the development of participatory processes in architecture. However, they also mention that these processes are time-consuming and that there is a lack of opportunity to practice them for various reasons.

Given the results of the interviews, we concluded that there are few cases in the practice of participatory processes in collective housing, and when they exist, they are limited to a low level of participation.

Although architects support participation, they are reluctant to let end-users create design proposals, arguing that this is the architect's role and that the community does not have the knowledge to do so. Even so, the foreigner-interviewed professionals are more open to including end users through collaboration in the creation and production of the project.

It was also observed that digital technologies are still little used in participatory processes, especially for end users to produce their own proposals. Even so, professionals highlight the potential of using digital tools, as they improve the perception of space by non-specialists in architecture, consequently allowing them to better understand regulatory possibilities and limitations.

We conclude that, despite being a complex process, professionals consider participation necessary and digital technologies beneficial for participatory processes. It would be positive to

use a digital tool that allows end-users to create their own proposals while the architects, the holders of the technical knowledge, define the limitations and design possibilities.

## Interviews with housing cooperatives

Representatives of ten Portuguese institutions that promoted the design and construction of collective housing between 1970 and 2020 were interviewed. The typologies of collective housing projects these institutions promote are single-house residential neighbourhoods and apartment buildings. The areas of activity of these institutions are Lisbon, Setúbal, and Azores, and they operate mainly for middle-low classes.

The representatives of housing cooperatives identified four phases of the housing definition and construction process: (1) land acquisition, which is generally carried out before members are registered; (2) the definition of the project, which is done with the participation of the cooperators by half of the interviewed cooperatives; (3) meetings with cooperators, carried out by half of cooperatives for participation, while in other cases were just to show them the project; and, finally (4) the construction phase, where adjustments and choice of finishes can be made.

In most cases (75%), the cooperatives' representatives asked for the cooperators' participation, even though this participation took place essentially in initial stages, such as the definition of requirements, and more advanced stages that do not influence the organisation of the space, such as the choice of materials during construction. The active contribution to the design of the proposals was limited.

The project was shown to the cooperators through technical and simplified floorplans, and visits to the construction site were made. Decisions were taken in some cases individually through forms regarding the choice of finishes and, in most cases, collectively by oral votes to decide on general aspects of the dwellings.

The cooperatives reported that the cooperators were motivated but showed no interest in participating more actively in the housing definition phase. Even so, they are convinced that most cooperators were satisfied with the result of the housing designs.

## Interviews with inhabitants of housing cooperatives

Twenty inhabitants of eight housing cooperatives were interviewed. These interviews were carried out with people who were the first tenants of the houses, as they were involved in some way in the design and construction process.

Although neither age nor gender were controlled, the interviewees ranged from 25 to 77 years old. They joined housing cooperatives when they were between 25 and 45 years old. Respondents have different levels of education and relationship with digital technologies. More than half of them (65%) had never had contact with architectural projects before their experience in the housing cooperative.

The interviews showed that inhabitants were not usually involved in the design phase of participatory processes carried out by housing cooperatives. When they were involved, they enjoyed having contact with the design and showed interest in participating more actively in the design definition. They would like to have explored design possibilities. Also, they showed no difficulties interpreting technical drawings when complemented with other less abstract ways of representing the project, such as perspectives of spaces.

With the last part of the interviews with inhabitants, we realised that they would like to participate in different phases of the housing definition process, essentially defining the quantity and dimensions of spaces. They would like to make decisions by interacting with predefined options, dragging them onto the design, or clicking on them to make choices that

automatically appear in the design. In terms of visualisation, 3D digital models and humanised floorplans stand out, and, finally, they chose the computer as the device they feel most comfortable using for defining their houses.

## Definition of the system–steps 2, 3 and 4

Although the user requirements (step 2) are explained in detail in [62] we mention them in this paper for context. The user requirements include that inhabitants should have direct contact with the design to make decisions about their houses, and the system should provide immediate feedback for the inhabitants to see the results of their choices in real time. The interaction with the interface should not require technical knowledge, and, thus a viable base design should be provided along with predefined options for the inhabitants to choose from. Another requirement is that the interface should provide diverse and realistic perspectives to enhance the understanding of non-specialists in architecture.

Based on the user requirements extracted from the interviews, we considered three different user profiles (the architect, the inhabitant (assisted by the system), and the system itself) to perform tasks along different phases (Fig 2). The design tasks identified in Fig 2 are the system features (step 3) and, thus, are explained in this paper along with the explanation of the interface definition (step 4) as they are directly related. The architect's task is to define the general characteristics of the building and construction process. Such characteristics include the number of floors, height, shared spaces, housing units shape configuration and distribution, energy efficiency of the building, type of construction system, etc. The conditions that will assist inhabitants' decisions regarding the interior of their houses in subsequent phases are design rules defined by the architect. Such conditions are based on their knowledge and experience,

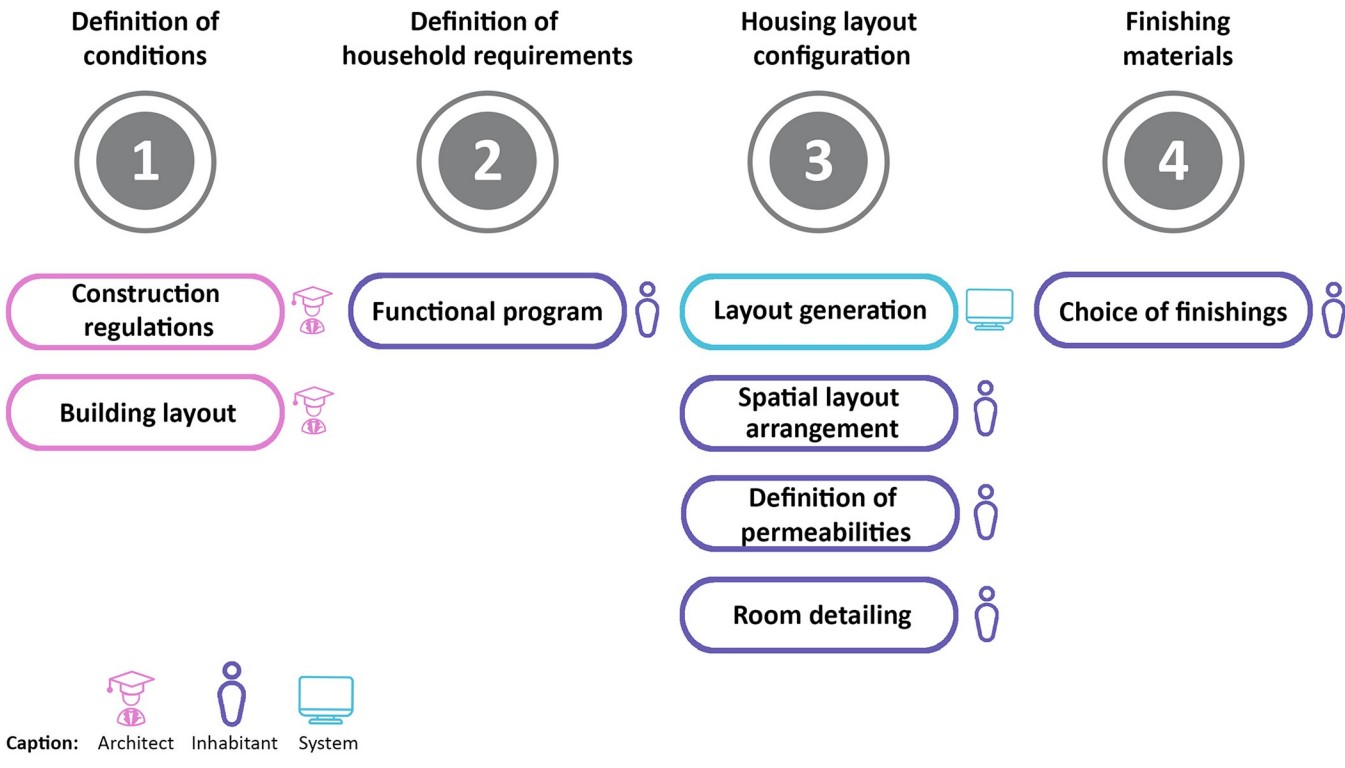

**Fig 2. Phases (1, 2, 3, 4) and tasks of the housing definition process.**

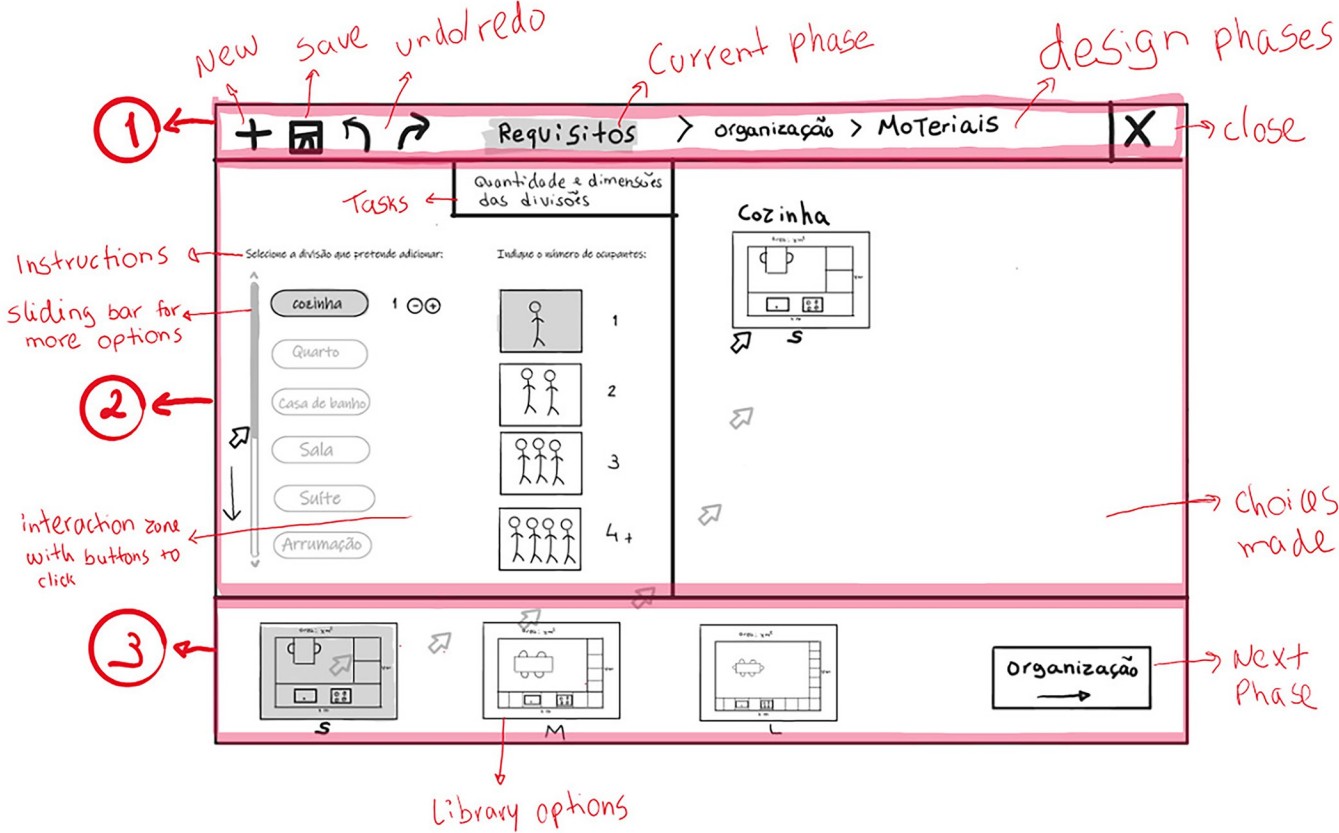

**Fig 3. Sketch of the user interface showing the task "Functional program" ("Quantity and dimensions of rooms"/ "Quantidade e dimensões das divisões") within the definition of household requirements ("Requirements"/"Requisitos") phase.** The three main areas are highlighted: 1- top toolbar; 2- main screen; 3- library bar.

as in Habraken's Supports method. The regulatory requirements are met through the generative system, which acts as a design critic, while the interface acts as an intermediary between end-users and the generative system fed by the architect.

In this research, we focused on providing a user interface for the tasks performed by inhabitants rather than for the tasks of the architect. As a general requirement, our interface must be easy to use, without the need for any technical knowledge from the user, and assist the user throughout the process. Thus, we propose that the design tool is based on a generative design system that automatically generates different viable housing solutions according to user preferences and presents them for users to choose and customise their house layouts. As mentioned before, this research focuses on the graphical user interface for the interaction with the final users (inhabitants), and therefore, we did not develop new research on the development of a generative design system, on the architect's use of such a generative design system and on the system interface for the architect.

Our Graphical User Interface (GUI) (Fig 3) contains three main areas: 1) a top horizontal toolbar with the design phases and generic buttons (e.g., save, undo/redo, close); 2) the main screen, divided into two sub-screens; and 3) a bottom horizontal toolbar corresponding to the library where the predefined options are displayed.

The design phases are tracked in the top toolbar, and the current phase is highlighted to ensure the user's awareness of the process. In this paper, we use technical terminology to

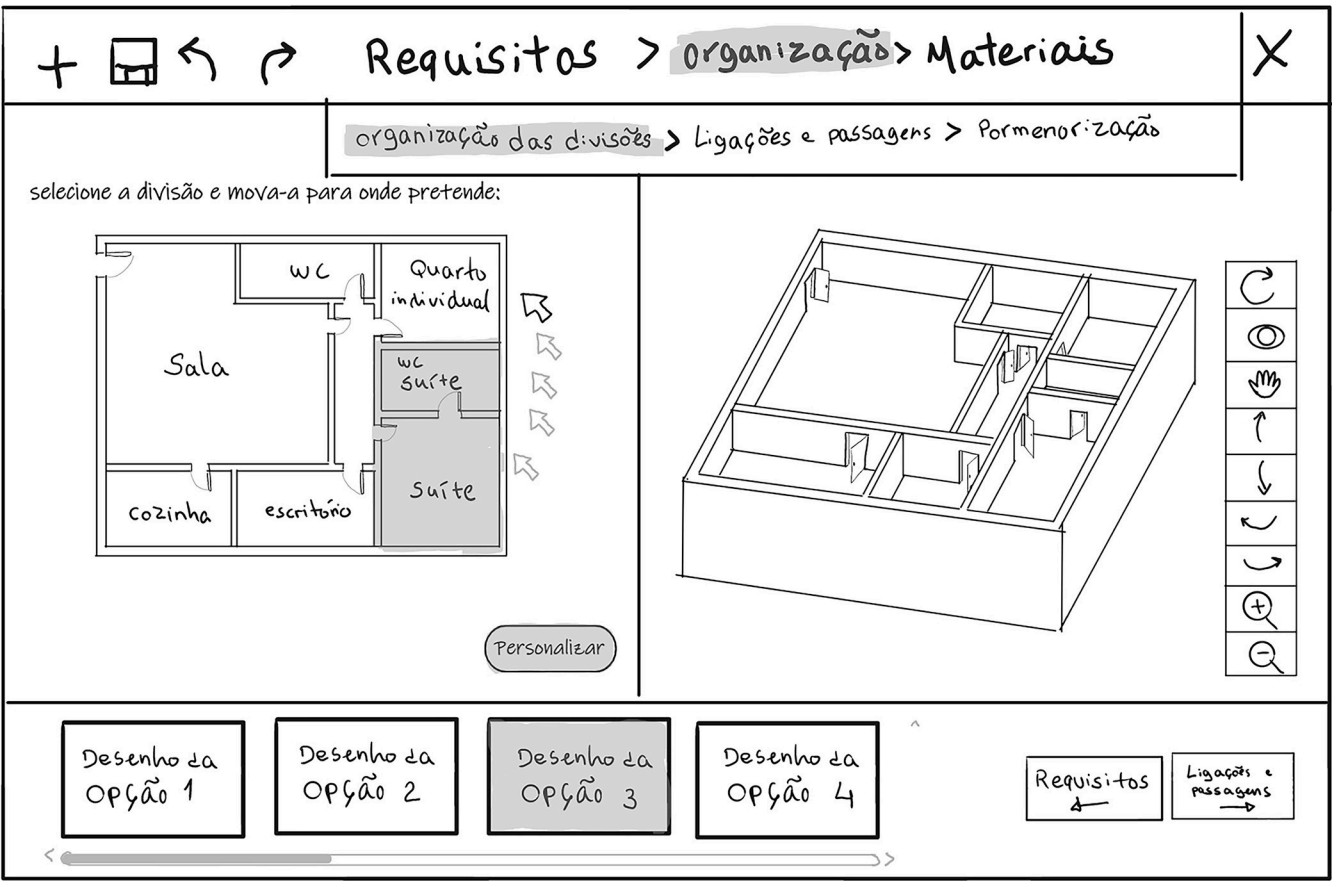

**Fig 4. Sketch of the user interface showing the task "Spatial layout arrangement" ("*Room's organisation*"/ "*Organização das divisões*") within the housing layout configuration ("*Organisation*"/ "*Organização*") phase, with rooms being dragged in the floorplan (left sub-screen) and the 3D representation on the right sub-screen.**

identify the design phases and tasks However, we used common terminology in the interface to respect the second Nielsen's heuristic [63] (Match between system and the real world) by using the user's language. On the right corner of the bottom toolbar, there are also buttons for the user to change to the next and previous phases (e.g., Fig 4). Throughout the entire design process, the interface provides instruction sentences to guide the user on what to do at each phase. These instructions appear on the main screen (Fig 3) and in the library, according to where the user needs to interact to perform the next step of the task. The main screen has right and left sub-screens, which display different features according to the task. The interaction in this workspace is made by clicking buttons and dragging icons from the library.

In the "Definition of the household requirements" ("Requirements"/ "Requisitos") phase (Fig 3), the user performs the task "Functional program" ("Quantity and dimensions of rooms"/ "Quantidade e dimensões das divisões"). In this task, they interact with the left sub-screen to choose the rooms and the number of occupants. If the user wants a room that is not on the list, the user can add such a customised room. For each room selected, the system shows size options (S, M, L). In the remaining phases, where there is already a housing layout to work on, the left sub-screen shows the floor plan of the house layout, and the right sub-screen displays a 3D representation (Fig 4). When the user changes the design, the system automatically refreshes both representations, and the user sees the results of their choices.

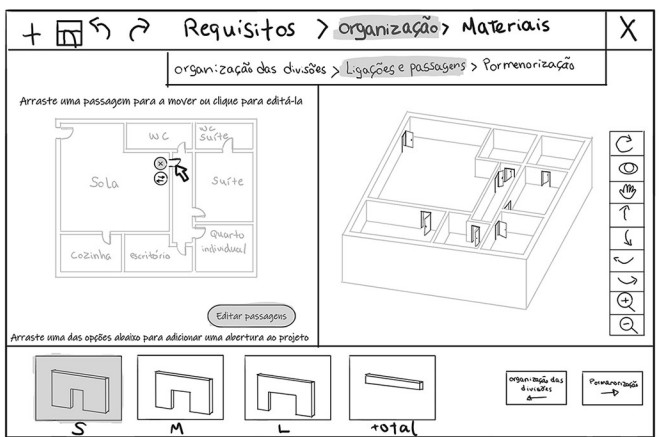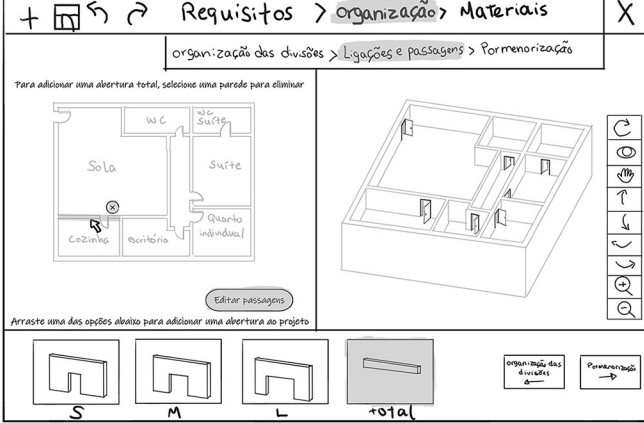

**Fig 5. Sketches of the user interface showing the task "Definition of permeabilities" ("*Connections and passages*"/"*Ligações e passagens*") within the housing layout configuration ("*Organization*"/ "*Organização*") phase.** Left: A door is selected, and a pop-up menu with options to change or delete it is visible. Right: The library option "total" is selected, and a wall is selected to be deleted.

In the "Housing layout configuration" ("Organization"/ "Organização") phase, the inhabitant performs three tasks: "Spatial layout arrangement", "Definition of permeabilities", and "Room detailing". In the "Spatial layout arrangement" task ("Room's organization"/ "Organização das divisões") (Fig 4), the user starts by choosing one of the layout options available in the library. The system generates these options based on the requirements defined in the previous task. After choosing the layout, the user can also change the position of the rooms by dragging them on the floorplan (left sub-screen).

To edit the connections between rooms, in the task of "Definition of permeabilities" ("Connections and passages"/"Ligações e passagens") (Fig 5), the user can click on each door to remove it, drag to reposition it or change the dimension of the opening choosing them from the library. If the user wants to connect two rooms by removing an entire wall (not structural or those related to the dwelling's infrastructure), the user selects the option "total" in the library and then selects the wall they want to remove.

In the "Room detailing" ("Detailing"/ "Pormenorização") task (Fig 6), the selectable rooms become highlighted while hovering the cursor. The user selects in the floorplan the kitchen or the bathroom they want to define. The system presents the library's different options, which the user can drag to the design.

In the "Finishing materials" ("Materials"/ "Materiais") phase, the user performs the task "Choice of finishings" ("Choice of materials"/ "Escolha de materiais"). The user selects the button corresponding to the room of the house they want to assign a material. The system presents the types of materials that can be dragged from the library. Fig 7 shows an example of wooden flooring being assigned to the living room floor. The system then calculates the budget based on room areas and chosen materials.

## Prototyping and evaluation–steps 5 to 10

### Formative evaluation using a low-fidelity prototype–step 5

Formative tests were carried out using a paper prototype as the second iteration of the tool development. This evaluation aimed to identify problems and opportunities for improving the interface. Tasks were defined for participants to use the system, and at the end, they answered

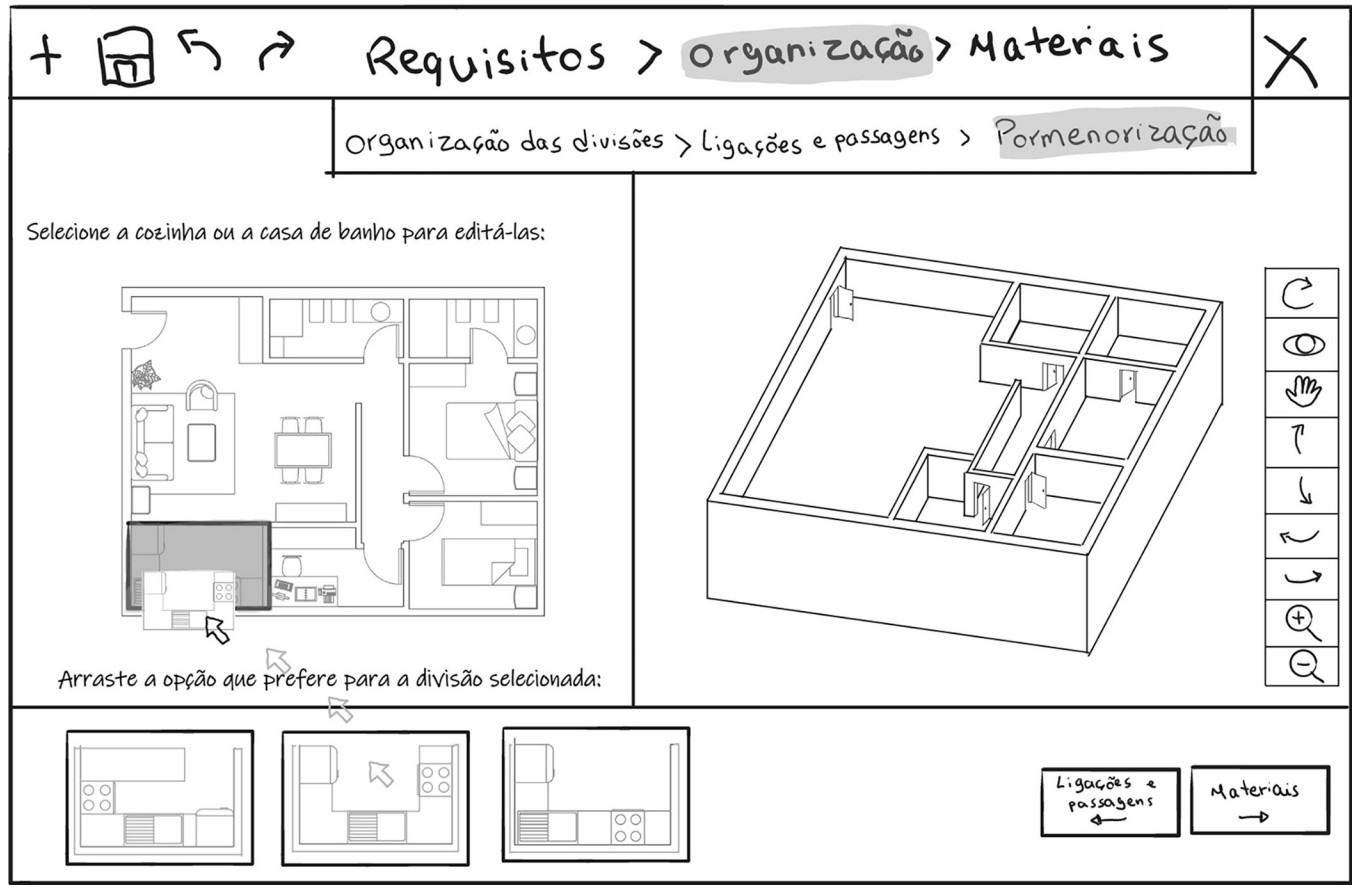

**Fig 6. Sketch of the user interface showing the task"Room detailing" ("*Detailing*"/ "*Pormenorização*") within the housing layout configuration ("*Organization*"/ "*Organização*") phase, with the kitchen selected and being replaced for one library option.**

three questionnaires to assess the usability of the system and their satisfaction. The participants also identified positive and negative aspects of the interface in a narrative way.

The tests were conducted with five participants, all inhabitants of the housing cooperatives that participated in the interviews done in step 1. Literature shows that this number of participants is enough to identify 85% of the problems of an interface [48]. Participants were between 25 and 45 years old, and gender was not controlled. The age range was identified during the interviews as potential users of such a design tool. Indeed, this is the age range in which people join housing cooperatives and are more willing to take part in the definition process of their houses.

A paper prototype was created representing the general layout of the interface and paper pieces corresponding to the buttons, the library's options, and the housing design's 2D and 3D representation. These pieces were attached to the interface layout and could be moved or replaced during the tasks' performance to simulate the system feedback to participants' actions (Fig 8).

Participants were asked to follow a script with various tasks. Since the prototype was only developed for one specific scenario and not for all possible scenarios, participants were asked to imagine that the indicated choices would be their preferences.

The results are described in [64] and allowed us to identify opportunities for improvement. In addition to some minor adjustments, the need for improvements is related to the task of

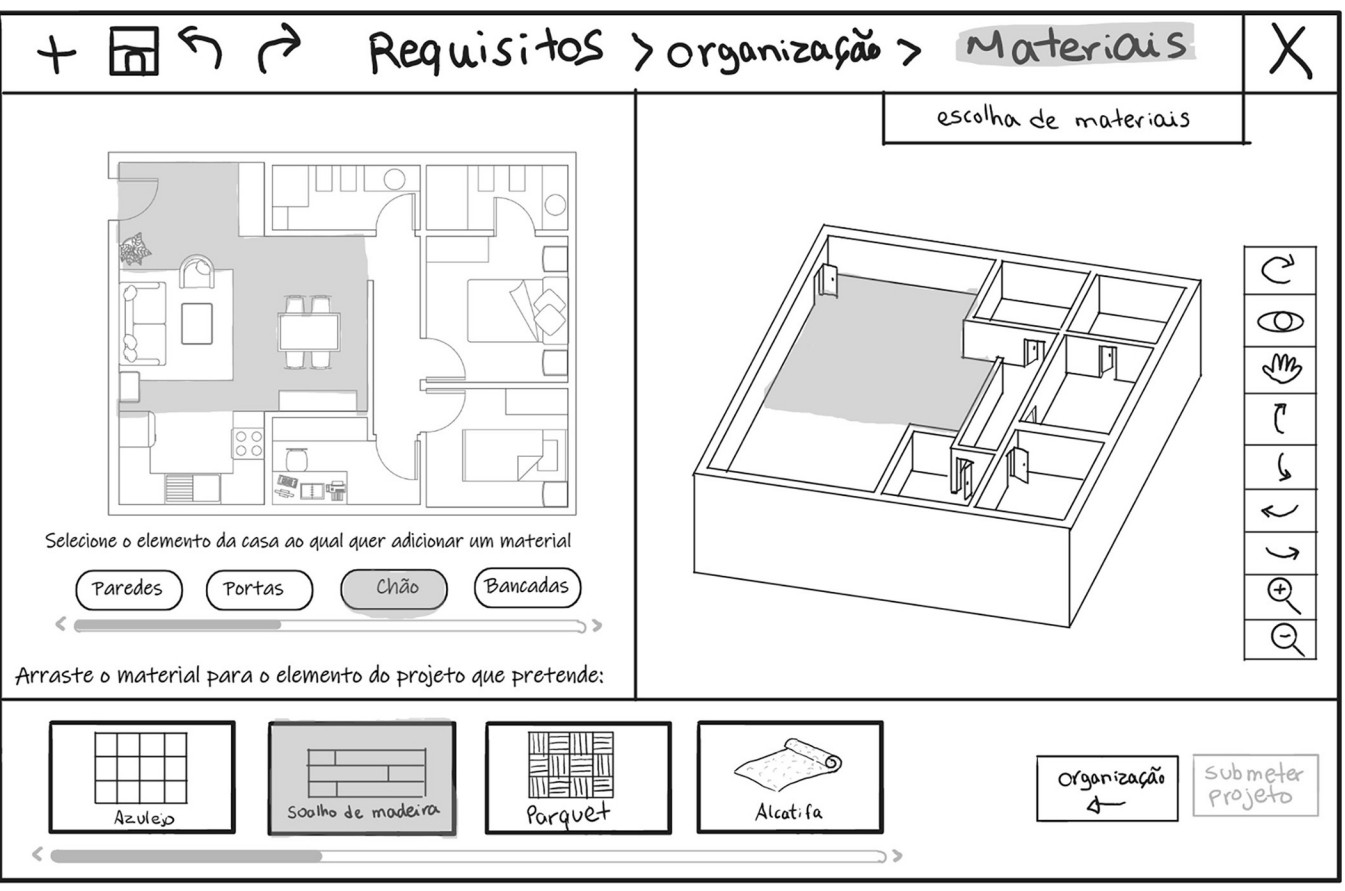

**Fig 7. Sketch of the user interface showing the task "Choice of finishings" ("*Choice of materials*"/"*Escolha de materiais*"), within the finishing materials ("*Materials*"/"*Materiais*") phase, with a wooden flooring assigned to the living room floor.**

defining permeabilities, i.e., connections between rooms. We also observed that dragging icons from the library was not intuitive, as the participants tended to click on options instead of dragging them.

## Refinement of the interface–step 6

A high-fidelity functional prototype was created using Figma [51], a codeless interface design online tool. A demonstration video using the prototype is available at [65]. This prototype accommodated the changes considered necessary after analysing the results of the tests performed with the paper prototype. The interface was also developed for the phases before choosing the housing requirements, such as the login phase and choosing the apartment.

One of the changes was the type of interaction, which used to be dragging options from the library to the working area. It was changed to just clicking on the option, and it automatically appears in the project. The drag-and-drop interaction is maintained in some other cases where dragging is more intuitive.

Another significant change was in the permeabilities task (Fig 9). During the first tests, we noticed the interaction was not intuitive for the users. We also realised that the option of eliminating doors would not make sense, as rooms always have a passage. Therefore, we removed this option and kept only the options to change the size of the openings and reposition doors, keeping the buttons visible without any pop-up menu. If it does not affect structure or

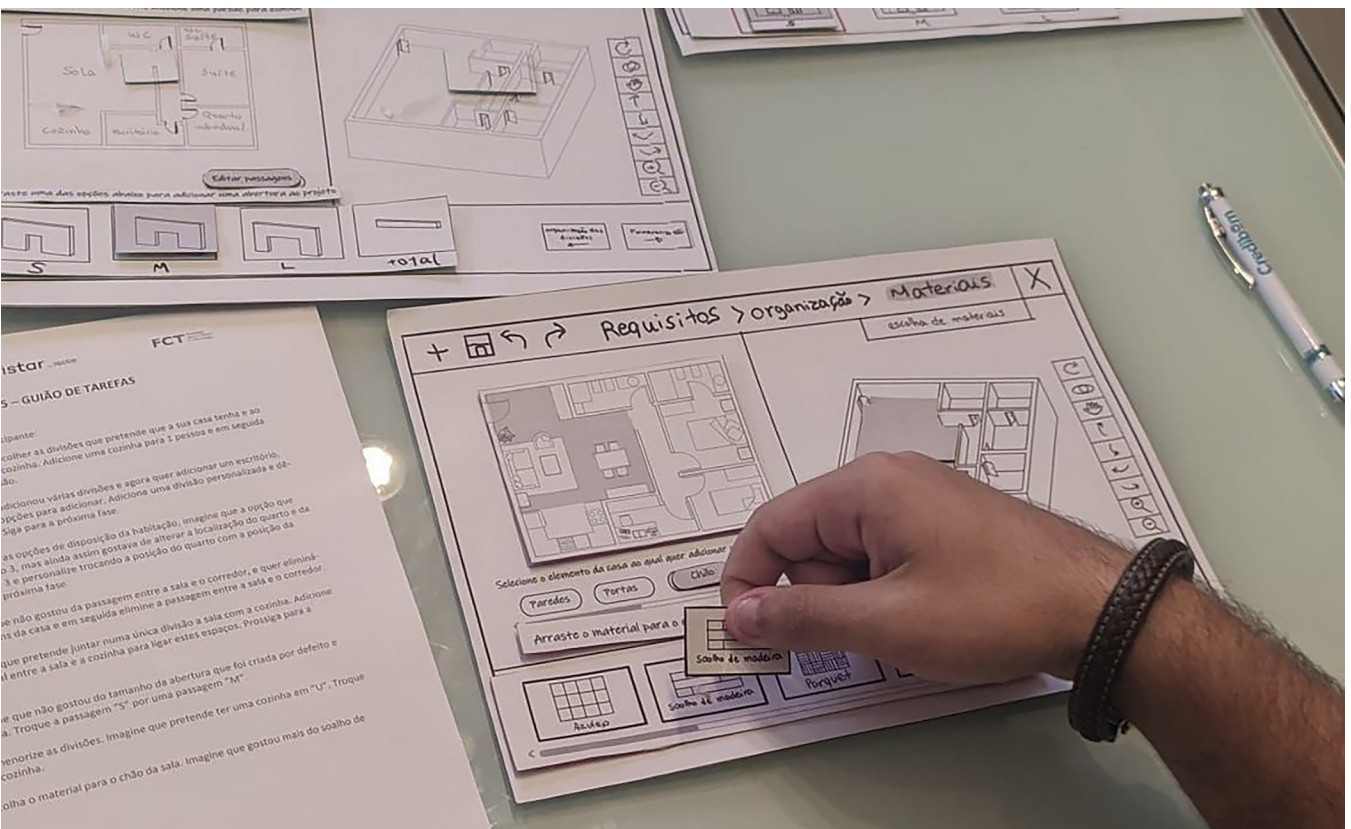

**Fig 8. Paper prototype with pieces being moved during the formative tests.**

infrastructure, walls can still be removed to connect rooms by pressing the button to edit walls and clicking on the wall to remove them.

The task of choosing materials was also further developed. We added a pop-up window with different options for choosing each material with associated costs and further developed the interface by showing the selected materials' budget (Fig 10). Also, as the type of interaction

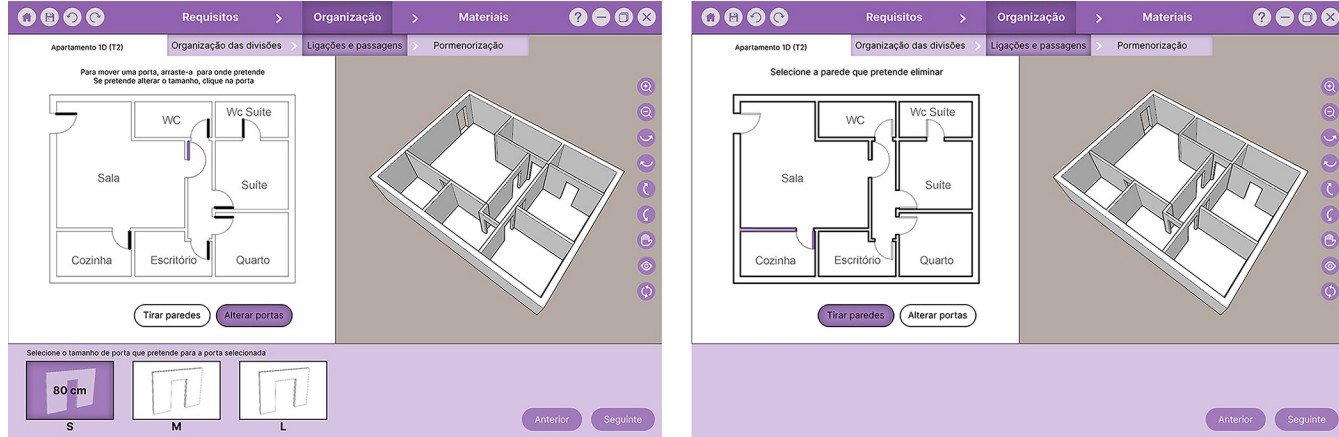

**Fig 9. Digital prototype interface with the changes made to the permeabilities task.** Left: "*Editing doors*" button activated and a repositioned door with *S* size selected; Right: "*Remove walls*" button activated and the wall selected to be removed.

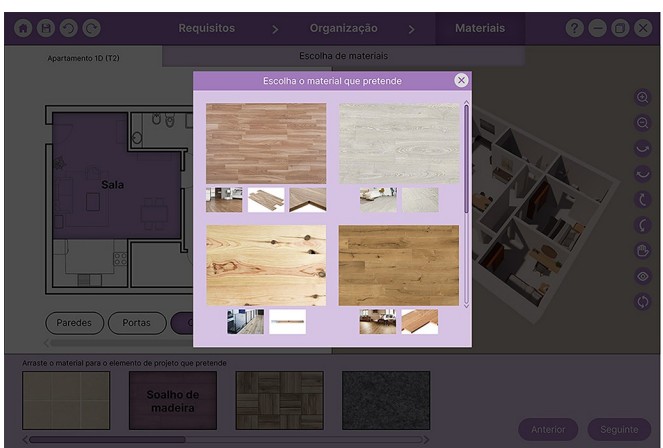 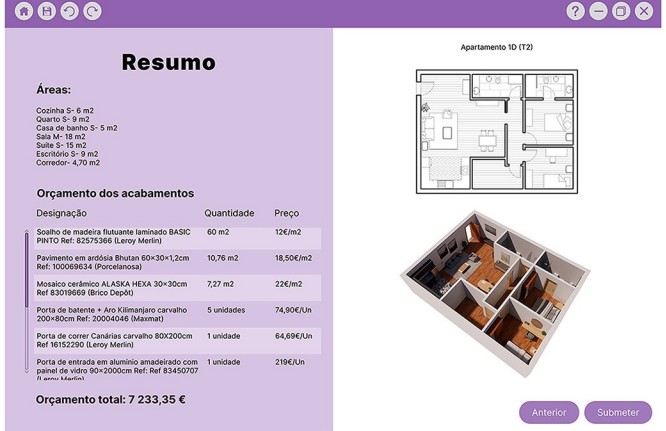

**Fig 10. Digital prototype interface of the "choice of materials" task.** Left: Window with different types of wooden flooring to assign to the selected room (living room); Right: Overall view showing the finishes budget.

was changed from drag and drop to click, we added the feature to first choose the room where the material is intended to be assigned.

After these significant changes and other minor adjustments to the interface, the high-fidelity prototype was tested with experts in user interface (UI) and user experience (UX). We also tested it with inhabitants (potential users). The goal of these two tests was to evaluate the usability of the system. We also gave the system to be used by a group of architects to obtain from them their perception regarding the use of such a tool for the co-design process. The study was conducted in an indoor environment with controlled noise and lighting for better concentration of the participants. The equipment used was a Surface Pro 8 with a mouse connected.

The study was conducted with five interface experts, thirty inhabitants and ten architects. Interface Design experts and architects were selected only for their professional practice, with age or gender not being considered in the choice of participants. In the case of the inhabitants, we selected the participants to keep the following characteristics balanced: age, gender, and education level (compulsory or higher education). The selection method was *snowball sampling* since the requirements were not complex, with the only mandatory requirement being between 25 and 45 years old (age range identified as potential users of the tool during the interviews).

Similarly to the tests made with the paper prototype, all the participants performed predefined tasks to interact with the system in different phases to try different features. In the case of interface experts, the goal was to identify usability problems that users could face and to correct them before carrying out the final tests with inhabitants and the discussion with architects. These problems were identified and categorised according to their frequency and degree of severity to define an order of priority in their resolution.

The tests with inhabitants intended to evaluate the system's usability and user-friendliness. Therefore, after experiencing the functionalities, they answered three questionnaires (SEQ-Single Ease Question, SUS- System Usability Scale, and GUI- Graphical User Interface). They were also asked to identify three best and three worst aspects of the system. We also registered the success or failure of completing the tasks and the time it took to complete them. Regarding architects, we aimed to collect their perception, as experts in design, about the benefits and limitations of using this design tool by inhabitants and in the scope of a co-design process and definition of customised housing.

In the following sections, we report the results of tests with each of the three groups mentioned above.

## Heuristic evaluation with interface design experts–step 7

A heuristic evaluation was conducted with five interface design experts. They identified usability issues and rated them with severity levels according to Nielsen's scale [66] from 0 to 4: 0) I don't agree that this is a usability problem at all; 1) Cosmetic problem only: need not be fixed unless extra time is available on the project; 2) Minor usability problem: fixing this should be given low priority; 3) Major usability problem: important to fix, so should be given high priority; and 4) Usability catastrophe: imperative to fix this before product can be released. The experts identified a problem, pointed it out in the interface, described it, rated it, and proposed a solution.

The results of the heuristic evaluation made by each UX/UI expert are reported in S4 File. Experts identified 32 usability issues (Fig 11), ten of which were rated as 1 (cosmetic problem only), nine rated as 2 (minor usability problem), and thirteen rated as 3 (major usability problem). Six usability issues were identified by more than one expert: five are from level 3, and one is from level 2.

The Nielsen heuristics [63] these usability issues infringed on were identified. The more frequent are the Heuristics 1 (Visibility of system status), identified in seven issues, and 2 (Match between system and the real world), identified in eight issues. Also, the heuristics 3, 4, 5, 6, 7, and 8 were infringed, identified in two-to-four issues.

Five levels of priority were defined for solving the usability issues according to the degree of severity (1, 2, or 3) and the frequency with which they were identified by the experts (the same problem was identified by one, two, or three experts). As Table 1 shows, the most severe and frequent ones have the highest priority (P1), followed by the severe non-frequent ones (P2), the frequent non-severe ones (P3), the non-severe nor frequent ones (P4), and those that, because they are just aesthetic problems, do not need to be resolved (P5). P1 and P2 correspond to problems with severity level 3, P3 and P4 are severity level 2, and P5 are severity level 1.

Each usability issue is described in detail in the S3 File. A brief description of the most severe and frequent (P1) issues is given below:

**Library options' caption.** The library options are not identified by a caption, which makes it difficult for the user to identify them, who should not have to memorise them.

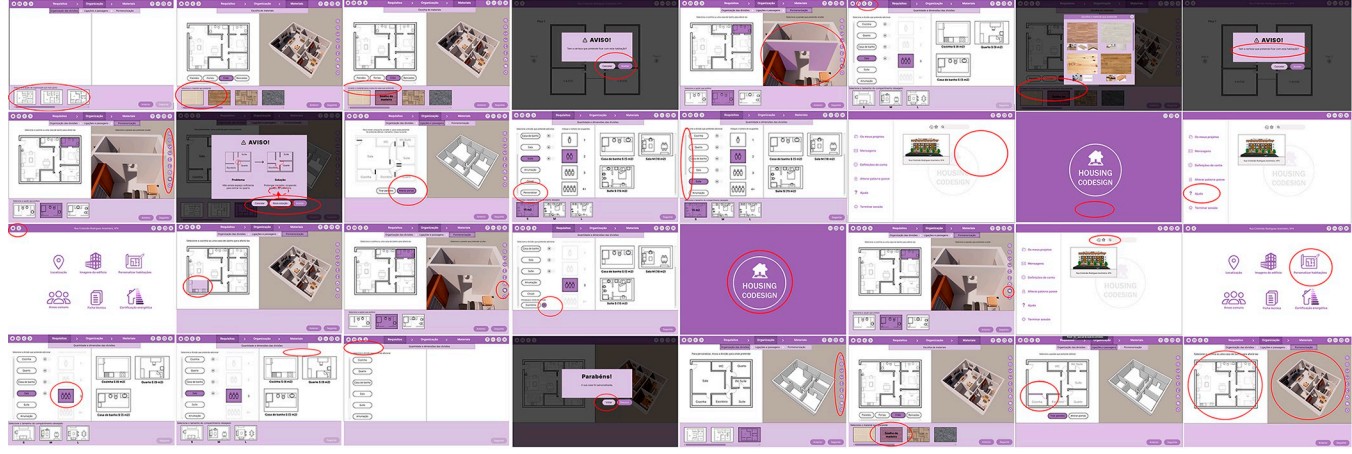

**Fig 11. Overview of all the usability issues identified by the experts, showing the interface and the location of the issue.**

**Table 1. List of usability issues identified by the experts.**

| Priority* | Issue | Severity** | Frequency*** |
|---|---|---|---|
| P1 | Library options' caption | 3 | 3 |
| | Order of associating the material with the element | 3 | 2 |
| | Drag the material | 3 | 2 |
| | Warnings | 3 | 2 |
| | Order of hiding walls in the 3D view | 3 | 2 |
| P2 | Undo/Redo | 3 | 1 |
| | Applying the material | 3 | 1 |
| | Irreversible action information | 3 | 1 |
| | 3D model menu option | 3 | 1 |
| | Accept/cancel new solution | 3 | 1 |
| | Order of changing the doors | 3 | 1 |
| | "Customise" button | 3 | 1 |
| | Scroll | 3 | 1 |
| P3 | "My projects" | 2 | 2 |
| P4 | Start | 2 | 1 |
| | Help | 2 | 1 |
| | Home and Back buttons | 2 | 1 |
| | Identification of rooms on the floorplan | 2 | 1 |
| | "Hide" button denomination | 2 | 1 |
| | Cancel button on the customise room option | 2 | 1 |
| | Logo | 2 | 1 |
| | "Hide" icon | 2 | 1 |
| P5 | Icons in "my projects" interface | 1 | 1 |
| | "My home" | 1 | 1 |
| | Number of occupants | 1 | 1 |
| | "Choices made" identification | 1 | 1 |
| | Apartment identification | 1 | 1 |
| | "Continue customising" button | 1 | 1 |
| | 3D orthographic views | 1 | 1 |
| | Materials denomination | 1 | 1 |
| | Delete walls | 1 | 1 |
| | Windows | 1 | 1 |

*_Priority_: P1 = more frequent and severe; P2 = severe non-frequent; P3 = frequent non-severe; P4 = non-severe nor frequent; P5 = Just aesthetic problem.

**_Severity_: 3 = major usability issue, 2 = minor usability issue, 1 = cosmetic problem only.

***_Frequency_: 3 = identified by three experts; 2 = identified by two experts; 1 = identified by one expert.

**Order of associating the material with the element.** Since the task is to choose a material, when the material is selected, the user considers the task complete, forgetting to associate it with a room. Thus, the steps must be done in reverse order.

**Drag the material.** The instruction to the user to drag the material to the desired room/element, despite pulsing, does not stand out (when solving the previous issue, it is no longer necessary to drag the material and therefore the pulsing sentence can be deleted).

**Warnings.** When warnings about important actions appear, the highlighted button should be the "cancel" button to prevent the user from doing actions they do not intend to by clicking the button without paying attention to its function.

**Order of hiding walls in the 3D view.** The walls can be hidden in the 3D menu to better visualise the rooms. However, the system should allow selecting a wall and then giving the instruction to hide it and vice versa because different users can act differently.

Other usability issues include the order of changing permeabilities, the need to change buttons or icons, indicating that certain actions are irreversible, etc.

## Second interface refinement–step 8

A new interface refinement was made based on the priority list, which resulted from the consolidation of the issues identified by the experts. The issues were solved by order of priority, and their resolution followed the experts' suggestions during the heuristic evaluation.

Only seven of the 32 issues identified were not solved. Almost all were from the lowest priority levels (P4 and P5). Four issues were categorised as cosmetic issues that do not need to be solved (severity level 1) and for this reason they were not solved. Such issues are: "3D orthographic views", "Materials denomination", "Delete walls", and "windows".

Two issues ("Logo" and "Hide icon") were categorised as minor usability issues (severity level 2). They were not solved because we considered that these issues do not interfere with the interface's usability, as other features guarantee the usability (e.g. the *Hide* icon was identified as an icon that does not immediately refer to its purpose, however, while hovering the cursor, the denomination of the button appears, so the users know the button's purpose even if they do not recognise the icon).

The "Scroll" issue, identified as severe but non-frequent (P2), was also not solved. It consists of hidden features that are accessible through a scroll bar. Scrolling should be avoided as all the necessary information should be visible on the screen, even if it implies dividing the features across more screens. We decided not to solve it because, although classified as severe (level 3), it was only identified by one expert. The resolution of such an issue would imply a significant intervention in the interface, which would significantly impact the research progress.

This interface refinement allowed us to improve our high-fidelity prototype, test it with potential end-users, and gather information about the interface's usability and users' satisfaction. Architects also used the prototype to discuss the interface's usefulness in a co-design process for housing customisation. The following sections report the results of the last iteration which includes the evaluation with inhabitants and the discussion with architects.

## Summative evaluation with inhabitants–step 9

Thirty potential users participated in the summative evaluation using the refined high-fidelity prototype. The inhabitants were not those who participated in the test with the low-fidelity prototype and never had contact with the interface so as not to compromise the results. 50% of them were men and 50% were women. Participants were between 25 and 45 years old (30% were 25–30 years old, 17% were 31–35, 13% were 36–40, and 40% were 41–45 years old). Regarding education, 43% had compulsory education, and 57% had higher education. They consider having a reasonable (37%), good (30%), and very good (33%) relationship with digital technologies, and 77% had never had contact with an architectural project.

The participants performed ten tasks to experience the different interface features along the design phases. The tasks script given to the participants to guide them in the interface experimentation is provided in S5 File. Task 1 consisted of logging in with the access data we provided. In task 2, participants were asked to choose the apartment that they would customise. In task 3, the participants start customising the house, by selecting a group of rooms and their sizes. Task 4 intended to add a customised room that was not on the list of rooms available to choose from. Task 5 aimed for the participants to choose one of the library options for the

housing layout and then customise it by swapping the position of two rooms. Tasks 6 and 7 are related to editing the connections between rooms. In task 6, the participants should drag a door to a new position, and in task 7, they should remove a wall to connect two rooms. Task 8 consisted of detailing the floorplan by choosing the kitchen and bathrooms layouts. In task 9, the participants should choose a flooring material and assign it to a room. Finally, task 10 consisted of finishing the process of customisation. For that, they should access the page that displays the summary of the choices and the budget information and then submit the project that would appear in the overall floorplan of the building.

During and after the interface experience, a set of questionnaires were applied. S6 File refers to the questionnaires the inhabitants answered to, and it includes a preliminary questionnaire, a Single Ease Question (SEQ) questionnaire, a System Usability Scale (SUS) questionnaire, and a Graphical User Interface (GUI) questionnaire. S7 File presents the results of the summative evaluation, describing the answers of each anonymised participant to each questionnaire. The inhabitants answered the SEQ questionnaire [67] by rating each task after completing it. They rated on a Likert scale from 1 to 7 (very difficult to very easy). Table 2 shows that the tasks were generally considered easy to perform, with averages above 5.97. Task 1 (logging in) was the highest rated, with 7 from all the participants. Task 2 (choosing the apartment in the building floorplan) had the lowest rate of 5.97 on average and a Standard Deviation (SD) of 1.13.

Each task consisted of a set of steps that should be performed to complete it. We registered the success or failure in carrying them out, the number of attempts made, and the time participants took to complete them successfully.

As Table 3 shows, all the participants completed all the tasks, although not all did it on the first attempt.

After performing all tasks, the inhabitants answered a System Usability Scale (SUS) questionnaire with ten questions to evaluate the system's usability. These predefined questions are alternately phrased positively and negatively. Each item is rated from 1 to 5 (totally disagree to totally agree). The SUS score is a number that represents the sum of each item score, which ranges from 1 to 4 [68]. The odd items (1,3,5,7 and 9) scores are calculated by subtracting 1 from its scale position (X-1). On the other hand, for even items (2, 4, 6, and 8), the score is 5 minus the scale position (5-X). If, e.g. item 1 was classified as 4 on the Likert scale, its score contribution would be 3 (4 minus 1), and if item 2 was classified as 1, its score contribution would be 4 (5 minus 1). The total SUS score ranges from 0 to 100, and it is calculated by multiplying the sum of each item's contributions by 2.5.

Table 4 shows the results of the SUS filled by the inhabitants. Each item score was calculated based on the average ratings from thirty inhabitants, and their sum was multiplied by 2.5. Each

**Table 2. Inhabitants SEQ questionnaire: results of each task.**

|  | Average | SD |
|---|---|---|
| Task 1 –Login | 7.00 | 0.00 |
| Task 2 –Choose the apartment | 5.97 | 1.13 |
| Task 3 –Selection of rooms and sizes | 6.70 | 0.53 |
| Task 4 –Addition of a customised room | 6.13 | 1.20 |
| Task 5 –Choose and customisation of a house layout | 6.10 | 0.96 |
| Task 6 –Change a door position | 6.67 | 0.61 |
| Task 7 –Remove of a wall | 6.63 | 0.61 |
| Task 8 –Choose kitchen and bathroom layout | 6.53 | 0.63 |
| Task 9 –Choose flooring material for a room | 6.10 | 1.06 |
| Task 10 –Submit the project | 6.77 | 0.63 |

**Table 3. Results of the task completeness by inhabitants.**

|  | Completed all the steps of the task (Number of participants) | Completed all the steps of the task in the first attempt (Number of participants) | Time average to complete the task (seconds) | Standard deviation |
|---|---|---|---|---|
| Task 1 | 30 (100%) | 30 (100%) | 25.7 | 6.3 |
| Task 2 | 30 (100%) | 24 (80%) | 36.0 | 13.1 |
| Task 3 | 30 (100%) | 30 (100%) | 28.6 | 12.4 |
| Task 4 | 30 (100%) | 21 (70%) | 23.6 | 14.7 |
| Task 5 | 30 (100%) | 29 (97%) | 24.9 | 9.5 |
| Task 6 | 30 (100%) | 30 (100%) | 8.3 | 6.5 |
| Task 7 | 30 (100%) | 17 (57%) | 12.4 | 7.5 |
| Task 8 | 30 (100%) | 30 (100%) | 26.2 | 11.2 |
| Task 9 | 30 (100%) | 21 (70%) | 22.9 | 9.5 |
| Task 10 | 30 (100%) | 30 (100%) | 19.7 | 9.2 |

item score contributed to a total SUS score of 93.17. A score of 80.3 or higher in SUS is identified in the literature as an excellent rating [68].

The inhabitants also filled out a Graphical User Interface (GUI) questionnaire, rating positively and negatively phrased statements from 1 to 5 (totally disagree to totally agree). The GUI questionnaire consisted of nine statements related to aspects of the interface.

As Table 5 shows, the positive statements had averages above 4.53, and the negative statements had averages up to 1.37 inclusive.

Finally, we asked inhabitants to list the system's three strong and weak aspects in their opinion. We grouped the answers according to identical aspects. Fig 12 represent the answers of inhabitants regarding both weaknesses and strengths. Twelve strong aspects were mentioned. 57% of the inhabitants said the tool is user-friendly, describing it as intuitive, simple, and easy to use. They also mentioned (53%) the fact that they can make their own choices in different design phases. Other aspects mentioned as positive were: i) the double visualisation mode as the floorplan and 3D model (37%); ii) the direct interaction with the design and seeing the results in real-time (30%); iii) the budget information (30%); iv) the predefined options to choose from (23%); v) the interface's graphical design being clean without too much information (23%); vi) the system warning if user's choices are not feasible or have relevant consequences, and suggests a solution (10%); vii) the potential to ease the dialogue between architect

**Table 4. Inhabitants SUS questionnaire: Results of each item and the total SUS score.**

|  | Average | Item score |
|---|---|---|
| Item 1- I think that I would like to use this system frequently | 4.60 | 3.60 |
| Item 2- I found the system unnecessarily complex | 1.27 | 3.73 |
| Item 3- I thought the system was easy to use | 4.73 | 3.73 |
| Item 4- I think that I would need the support of a technical person to be able to use this system | 1.20 | 3.80 |
| Item 5- I found the various functions in this system were well-integrated | 4.63 | 3.63 |
| Item 6- I thought there was too much inconsistency in this system | 1.27 | 3.73 |
| Item 7- I would imagine that most people would learn to use this system very quickly | 4.67 | 3.67 |
| Item 8- I found the system very cumbersome to use | 1.10 | 3.90 |
| Item 9- I felt very confident using the system | 4.57 | 3.57 |
| Item 10- I needed to learn a lot of things before I could get going with this system | 1.10 | 3.90 |
| Sum of item scores | - | 37.27 |
| SUS score (sum of item scores x 2,5) = 93.17 |  |  |

**Table 5. Results of the GUI questionnaire applied to inhabitants.**

|  | Average | SD |
|---|---|---|
| 1—I understood the names of the design phases. | 4.67 | 0.61 |
| 2—It's not easy to navigate through the menus | 1.13 | 0.35 |
| 3—I understood the menu options and instructions | 4.63 | 0.67 |
| 4—Menu icons and buttons do not clearly indicate their functionality | 1.37 | 0.56 |
| 5—Graphics and diagrams are easy to read and understand | 4.77 | 0.50 |
| 6—The screen graphical design is not well designed | 1.13 | 0.43 |
| 7—Information is well organised on the screens | 4.53 | 0.68 |
| 8—It's not easy to access the information I need | 1.13 | 0.35 |
| 9—The size of the buttons and words or statements is adequate | 4.73 | 0.64 |

and client, as it allows the inhabitants to translate the house they imagine into reality (7%); viii) the potential to be used in refurbishments (3%); ix) the order of the design phases is clear (3%); x) the pulsing sentences catch the attention (3%).

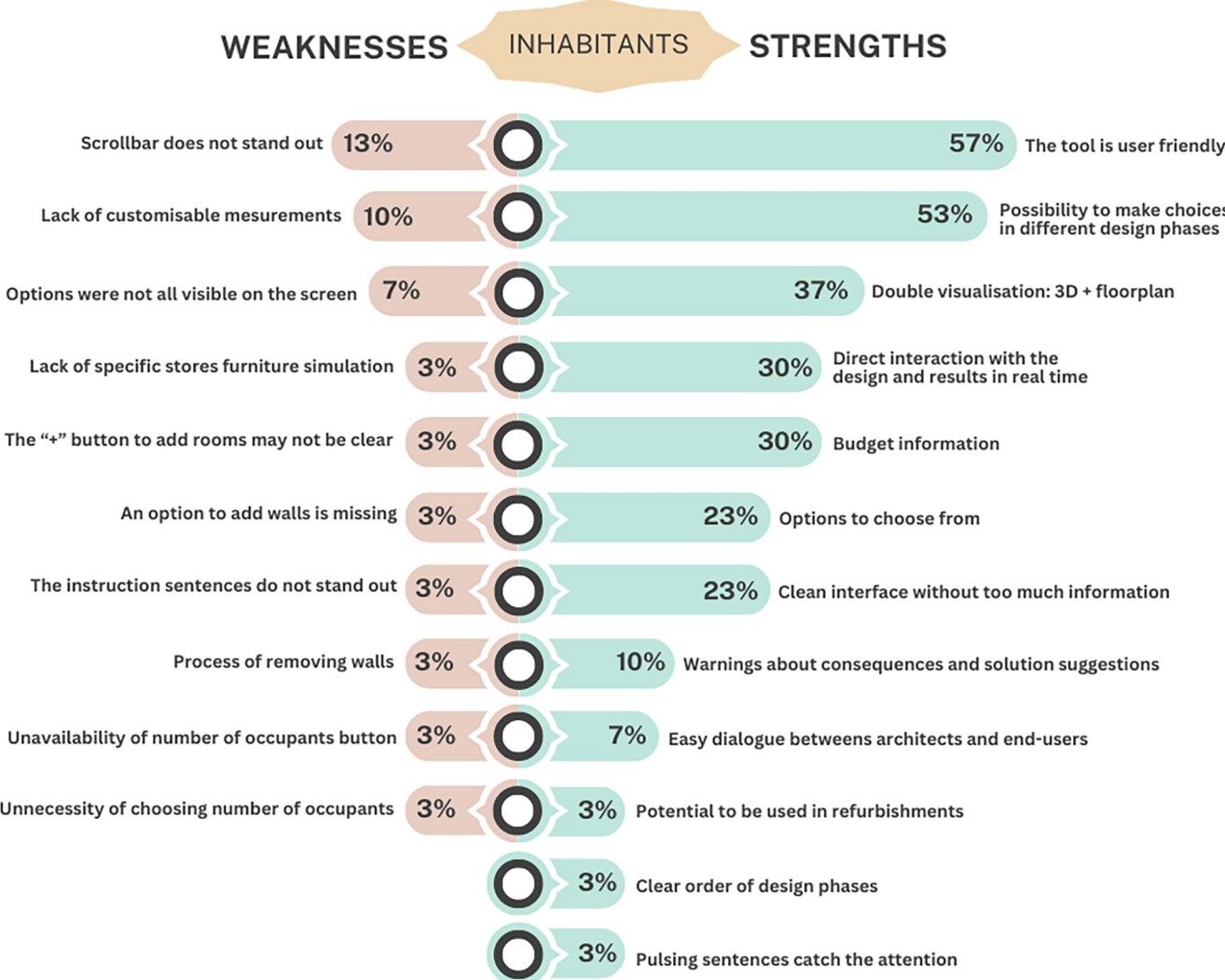

**Fig 12. Overview of all the weaknesses and strengths identified by the inhabitants.**

Regarding weak aspects, which are missing or could be improved, ten were mentioned. 13% of the participants said that the scrollbar in the definition of household requirements does not stand out. They also said they would like to be able to customise specific measurements, such as kitchen cabinets (10%). The fact that the options were not all visible on the screen (when defining the requirements), because they needed to scroll down the scrollbar to see more options, was also pointed out as an aspect to improve (7%). The remaining weak aspects were only mentioned by 3% (one person). Such aspects are: i) the system could simulate how the furniture of specific stores would look in the space; ii) the "+" to add rooms may not be clear to people who do not have much digital literacy; iii) an option to add walls is missing; iv) the instruction sentences do not stand out; v) removing walls should be done just by directly click on them without having to activate a button first; vi) when is not possible to choose the number of occupants this option should not appear; and finally, one person think vii) it is not necessary to choose the number of occupants of the household.

## Discussion with architects–step 10

Ten architects used the interface, performing the same ten tasks while playing the role of inhabitants to experience how they would participate in the definition of their own houses. Then, they gave their opinions from their expert viewpoint.

After performing all tasks, we asked architects to identify strong and weak aspects of the system. The results of the answers of each architect are provided in S8 File. As shown in Fig 13, a total of thirteen positive aspects were mentioned. The ones that stand out are: i) clear representation of the architecture through the floorplan and 3D model (identified by 70% of architects); ii) clarity, ease, and speed of use (60%); and iii) choices informed by the budget (60%). Other aspects mentioned include: iv) there is flexibility by allowing to choose between several options at different stages (30%); v) direct interaction with the project and results in real-time (20%); vi) sequence of tasks is correct (20%); vii) has the potential to be adapted to other participatory contexts (20%); viii) allows the design of customised houses because the inhabitants are the ones who make the choices (20%); ix) the interface's graphical aspect is clear (10%); x) can be used by people with little digital literacy (10%); xi) the ability to choose the number of occupants and room's size (10%); xii) does not require technical knowledge but still creates regulatory-appropriate solutions (10%); and xiii) makes inhabitants think about their priorities in managing the room's sizes (10%).

Regarding negative points, eleven aspects were identified regarding two subjects: (1) the interface and (2) the system itself. Regarding the interface, architects mentioned that: i) the instruction phrases, despite pulsing, do not stand out (40%); ii) the representation of site context is missing (10%); iii) the floorplan representation should be bigger (10%); iv) the buttons to edit doors and walls should be more visible (10%); and v) when choosing the apartment, there should be tabs identifying the building floors (10%). Regarding the system itself, they think that: i) it limits the architect's work since it only allows them to define the general characteristics of the building and shared spaces, but not the interior of the houses (20%); ii) requires from the inhabitant the ability to have full knowledge of space when choosing the rooms' size (20%); iii) is restrictive, as choices are conditioned only to the predefined options (20%); iv) only allows working on conventional architectural models, as the pre-defined options do not support projects with particular characteristics (20%); v) It can only be used when the end users are known before the construction (10%); and vi) the flexibility should not interfere with the budget (10%).

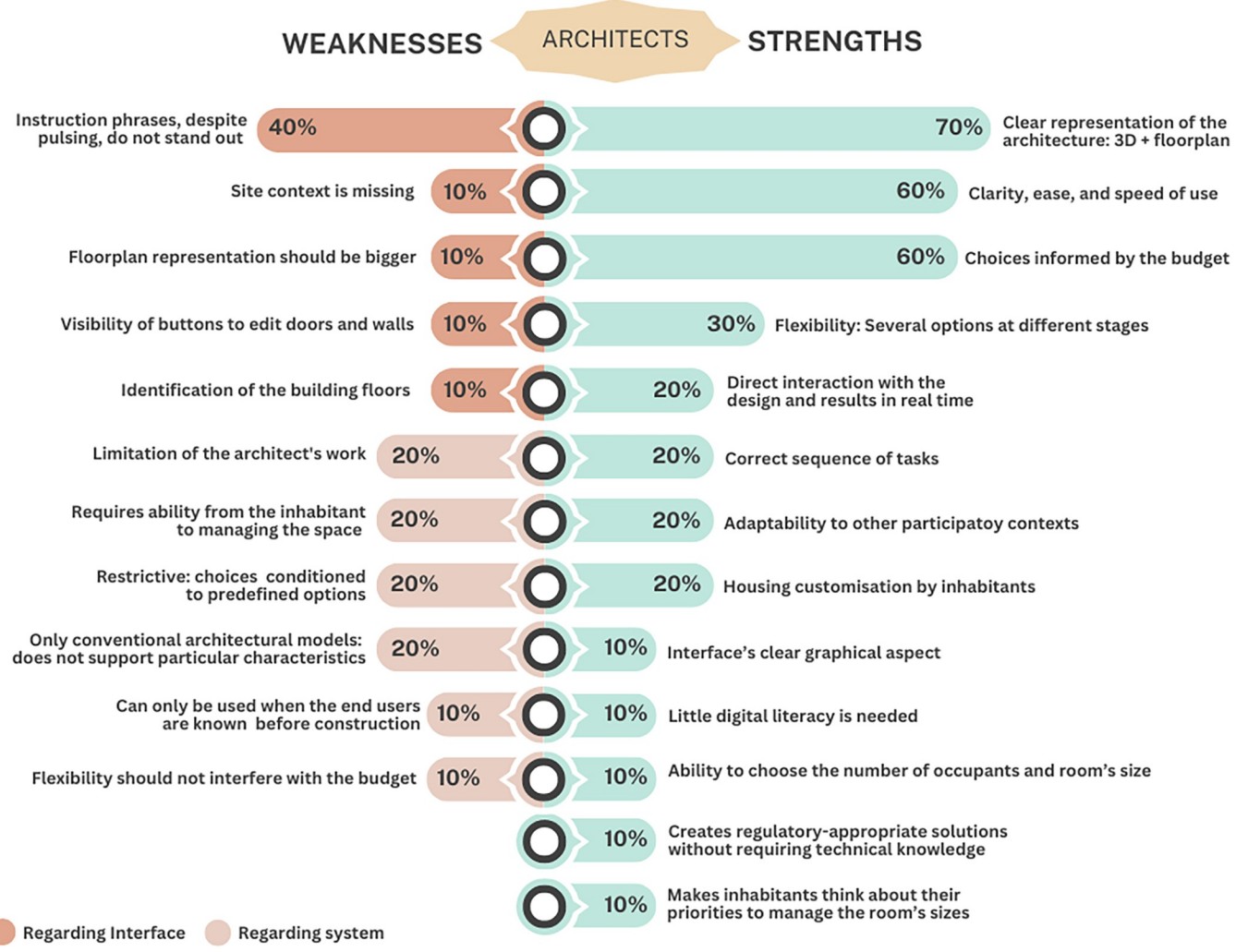

**Fig 13. Overview of all the weaknesses and strengths identified by the architects.**

## Discussion

The high ratings of the inhabitants' assessment of the interface show that the proposed system can be accessible to its target users and meets their needs. According to the results of the SEQ questionnaire, which assesses the ease of performing tasks, the tasks proved to be easy to perform since they were completed successfully by all the participants, although not all were completed in the first attempt. The difficulties that prevented some participants from carrying out some tasks on the first attempt are essentially related to the lack of visibility of some interface elements and the conditioning of the task to the script instructions. Still, the tasks were all completed. The task completeness records showed that the time taken to complete them varied, with some tasks taking longer than others, but all were completed within a reasonable time frame.

Regarding strengths and weaknesses, some aspects identified as strengths of the interface were identified by both groups (inhabitants and architects). Inhabitants highlighted aspects related to the interface itself (graphic appearance and user-friendliness both in terms of interaction and visualisation) and how the interface helps them to participate in the design process

(being able to make their own choices, having predefined options to choose from instead of creating from scratch, system warnings about feasibility, budget information, etc.). Architects highlighted that the interface is user-friendly, the clear representation of the architecture through the floorplan and 3D representation, and the budget information. Indeed, computer-aided design tools are used by designers to assist in their task of creating design solutions. However, their graphical user interfaces can be complex for non-designers who have not received training to manipulate them. Thus, we developed a graphical user interface to assist and guide inhabitants in a complex design decision-making process, by presenting options to choose from rather than asking them to manipulate design tools to create solutions from scratch. Based on the generative system, the interface presents the options and implications of choosing such options, acting as a design assistant. These aspects were considered in the interface design development to comply with the requirements derived from the interviews, and in fact, they were a positive aspect highlighted by the inhabitants during both the formative and summative evaluations.

We consider that using a User-Centred Design (UCD) methodology was paramount for this user-friendliness highlighted by both groups of participants. Using UCD to create graphical user interfaces in the housing co-design is not usual, as much as we could find in the literature. The development of some co-design systems presented in the literature (e.g. ModRule [44] and HOPLA-Home Planner [29]) involved testing their frameworks with different types of users. However, their authors tested them with architectural students acting as designers and end-users (who did not reflect the characteristics of the target users) or conducted an interface evaluation with real end-users using a prototype in an advanced phase. Although both produced paper prototypes to test their architectural design methods, we did not find evidence of end-user involvement in the interface design in preliminary phases following an iterative process, as in a user-centred design approach. Such an approach is essential to achieve a product that meets end-users' needs and satisfaction in graphical user interface development [48]. Inhabitants had difficulty mentioning something as a negative or missing aspect. Even so, they mentioned difficulties carrying out the tasks (e.g., interface elements that were not visible) and the lack of features they would like the interface to include (e.g., simulating furniture and customising measurements).

Regarding the negative aspects mentioned by the architects, despite saying that the system is adaptable to other contexts and pointing out flexibility as a positive aspect (as it allows choices to be made in various design phases), architects find the tool restrictive. They state that choices are conditioned by predefined options and that these options do not support projects with particular characteristics. Although this statement by architects is correct, the system for mass customisation of houses defined in this study focuses on middle and low-income households. Therefore, we propose that such projects have regular characteristics and that a couple of options are available for inhabitants to customise their houses. The system is thus not foreseen for design projects with particular characteristics, such as when a private client chooses an architect for a tailor-made design. Nevertheless, in the proposed interface, constraints are defined by the architects and the system embeds their knowledge and experience, including construction regulations. Capturing designers' knowledge in a generative design system allows users to explore various design options that respect with formalised design rules and construction regulations [29]. The interface acts as a design critic, as in Frazer's approach of Segals's method [34], or in HOPLA-Home Planner [29], by evaluating the implications of users' choices based on the design rules and presenting them possible solutions.

On the other hand, architects state that flexibility should not interfere with the budget. According to Khalili-Araghi and Kolarevic [28], flexibility for customisation does not sacrifice

efficiency and affordability if constraints are defined. The authors argue that these constraints can be standard predefined options that, when combined, allow for affordable customisation.

Architects also mentioned that the system can only be used in projects where it is known, before the construction, who the end-users will be so that they can participate in the design process. In fact, the tool was defined for a collective housing context, focusing on apartment projects for housing cooperatives. Still, this is just a context in which the users are previously known and can be applied to other situations, such as in the public initiative, with the relocation of specific communities, or private initiative in cases of single-family houses, in which the client directly contacts the architect or a built construction firm to design their house. By proposing such a co-design system, we also propose that in some social housing developments, the housing attribution system starts before the building construction so that the households can have a say in their future houses.

Architects consider that the system limits their role as they are not the ones to define the interior of the dwellings. The architect's role is not to create the design but to provide the framework and design rules for users to create customised designs. It's the inner logic rather than the external form. This discussion leads to the debate on authorship and stakeholders' roles in participatory and co-design processes. As discussed in the literature, the notion of authorship has been diluted with the development of collaborative digital tools and new ways of practising architecture, such as participatory design [69, 70]. In such a view, authorship is shared between the architect, who defines the conditions, and the end-user, who customises the design [69, 71]. The architect's role is to facilitate the process and give end-users the means to achieve a customised solution. As stated by Habraken [20], architects' mission is to feed the living environment to contribute to its spontaneous development. In the housing co-design system presented, the architect is the one who defines all the characteristics of the building, including the relationship with the geographical and cultural contexts, and also defines the various conditions and inhabitants' choice possibilities. These conditions are what will feed the generative system to present options to the inhabitants according to their choices, which means that the architect's work is highly necessary and valued.

Despite this, we can say that architects consider the presented system relevant to the practice of architecture involving end-users. They value the flexibility of choices so that the dwellings are truly customised. However, they fear being devalued because they do not understand the importance of their role in the process. For architects to value participatory and co-design processes, they must understand the importance of their participation, which is significant but different from what they are traditionally used to. Until architects gain awareness of how digital processes can be involved in the design they will resist this type of innovation and not recognise their relevance.

## Conclusion

In this paper, we start by arguing that the participation of end-users in the design process is a critical factor for achieving a house design that satisfies their aims. Digital technologies can facilitate informed decisions and engagement and, therefore, are relevant to participatory design processes. In this research, we developed a graphical user interface that assists inhabitants in co-designing their houses.

The prototype was tested with interface design experts, potential users and architects. The results showed that the interface was considered user-friendly and easy to use, with high ratings on the SEQ, SUS and GUI questionnaires.

The results of task performance and SEQ questionnaire showed that the architectural tasks provided by the system for housing customisation are easy to perform. Also, the results of SUS

and GUI questionnaires showed that the interface is user-friendly. We hypothesised that a digital co-design tool can assist inhabitants in the task of design for houses in an informed way, arguing that digital technologies improve the end-user understanding of space and have the potential to enhance their collaboration in the design process. We also aimed to identify the impact of the inhabitants using a digital design tool in the scope of a co-design process. We found, through the answers of positive and negative aspects, that the inhabitants feel satisfied using the interface as it allows them to collaborate on the design definition in a viable and informed way, achieving customised housing. The inhabitants referred to the interface as easy to use, and one of the aspects they highlighted as positive was the double visualisation mode. Besides, although the limitations discussed, the architects highlighted the potential of the digital tool by the clear representation of architecture and budget information, which empowers the end-users to make informed decisions. These results are aligned with the literature, which demonstrates the positive impact that 3D interactive and immersive visualisation has on the non-designers perception of space [2, 3, 72]. The potential to improve collaboration in an informed way is also related to the use of generative design technologies. Although this part is not currently implemented, we were able to simulate its operation using Figma and the script given to the participants of the tests, allowing the interface to inform the users of the results of their choices and present options according to their inputs. The participants of both groups also mentioned these aspects as positive. As mentioned in the literature by authors such as Kwiecinsky and Kolarevic, generative design, because it is based on pre-validated rules, is essential to instantly validate solutions created by non-designers and thus allow them to co-design in an informed way. Thus, we conclude that the inhabitants can easily interact with the architectural design of houses using a digital tool. Such results respond to the research question raised and validate the hypothesis presented as a basis for this study.

These findings lead us to conclude that the digital tool envisioned in this study can be a valuable addition to participatory processes in collective housing. It has the potential to improve end-users' satisfaction and enhance participation in their houses' design process since it allows for the customisation of housing while still accommodating construction regulations and budget constraints. This level of engagement facilitates the creation of designs tailored to meet the specific needs of inhabitants. By means of a digital co-design tool, architecture can respond to the users' diversity.

The innovation of this research lies in the fact that the interface was defined and tested with direct contact with potential users. We followed a well-established methodology (user-centred design) that is framed in the computational engineering sciences, but in this research, we applied it to solve an identified problem in the architectural field. Other works developed interface solutions for housing customisation, however, there is no evidence of their development with close contact with potential users.

However, a limitation of the study is that the tests were done using a small part of the envisioned system features since we used a prototype instead of a fully operational tool. The fact that the system was developed based on the requirements collected from a sample of only Portuguese participants is also considered a limitation of the research. Further development and testing with a broader audience would enrich the research and allow us to draw conclusions about the system's applicability in other geographic and cultural contexts. Future work also includes improving the visibility of certain elements of the interface and the full implementation of all features, coupling this interface design with a generative design part. We also suggest, as future research, the development of a graphical user interface for the architects, as well as the framework for them to introduce the design rules to feed the generative system. Such improvements can result in a fully operational tool that can be tested in a real co-design

environment and lead to the widespread use of such tools in architectural practice and within participatory processes for defining customised housing.

## Supporting information

**S1 File. Interview protocol used with the three groups of interviewees.**
(PDF)

**S2 File. Results of the interviews.**
(XLSX)

**S3 File. Description of each usability issue.**
(PDF)

**S4 File. Results of the heuristic evaluation.**
(XLSX)

**S5 File. Tasks script inhabitants followed in the summative evaluation.**
(PDF)

**S6 File. Questionnaires applied in the summative evaluation.**
(PDF)

**S7 File. Results of the summative evaluation.**
(XLSX)

**S8 File. Results of the discussion with architects.**
(XLSX)

## Acknowledgments

The authors would like to acknowledge every participant in the interviews and experiences conducted during the research. The manuscript is based on work presented and published at the Interfaces and Human-Computer Interactions (IHCI) International Conference in 2023.

## Author Contributions

**Conceptualization:** Micaela Raposo, Sara Eloy, Miguel Sales Dias.

**Data curation:** Micaela Raposo.

**Formal analysis:** Micaela Raposo.

**Investigation:** Micaela Raposo.

**Methodology:** Micaela Raposo, Sara Eloy, Miguel Sales Dias.

**Project administration:** Micaela Raposo.

**Resources:** Micaela Raposo, Sara Eloy, Miguel Sales Dias.

**Supervision:** Sara Eloy, Miguel Sales Dias.

**Visualization:** Micaela Raposo.

**Writing – original draft:** Micaela Raposo.

**Writing – review & editing:** Micaela Raposo, Sara Eloy, Miguel Sales Dias.

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
