## [Decision Letter · Decision Letter 0]

1 Aug 2024

PONE-D-24-25657Customised housing design: Bridging gaps through a digital tool for users’ collaboration.PLOS ONE

Dear Dr. Raposo,

Thank you for submitting your manuscript to PLOS ONE. After careful consideration, we feel that it has merit but does not fully meet PLOS ONE’s publication criteria as it currently stands. Therefore, we invite you to submit a revised version of the manuscript that addresses the points raised during the review process. The manuscript demonstrates an original contribution. However, it needs much more work on developing a critical literature review and on discussing how the outcomes contribute to participatory design and co-design. There are also improvements needed in the presentation of the steps leading to the design and prototyping of the user interface. I trust the authors will consider my comments below and the comments provided by the reviewer when revising the manuscript that requires significant improvements to be considered for publication. - The authors will ensure to provide a link to the complete dataset.

- The scope of the study needs to be more clearly articulated in the introduction and in the abstract and title. It appears that the focus of the manuscript is that of designing, developing, prototyping and testing a graphical user interface that facilitates end-user participation in co-designing buildings.

 -The framework needs to be enriched by including further reference to generative design tools and a discussion on the challenges when engaging users in participatory design.

 - In the methodology, a clearer justification for the choice of the sample could be provided by indicating the criteria used to select the professionals and housing cooperatives for the study. For example why were there 15 professionals from Portugal and 13 from other countries, how were the latter identified and selected for the study?

 - In the methodology, it would be helpful to include a diagram summarizing the 8 steps and how these feed into the development, prototyping, testing and evaluation of the user interface. It would be helpful to provide the interview protocols used with the three groups of interviewees.

 - The presentation of the manuscript needs to be improved. For example, the section titled: “Collection of user needs by interviews – step 1’ is somewhat confusing as some of the headings refer to “interview” outcomes and other subheading refer to the “steps” used to design the digital interface. It would be good to be consistent and refer to the steps in the design and use the findings from the interviews with the 3 groups of respondents to explain how these findings were used to inform each of the 8 steps. The section is too long and the authors could use sub-headings to explain the steps involved and then include a separate section tackling user perceptions and feedback. On p. 20, which step does ‘Evaluation of the high-fidelity prototype’ refer to ?

 - The discussion is very descriptive and though it provides useful practical tips on GUI and their applications, there is very limited (if any) discussion on how the study engages with and builds on relevant literature on the use of GUI in co-design and participatory design. The reviewer's comments below provide valuable insights on how this could be improved.

- The authors need to acknowledge that the manuscript is based on work presented and published at the 'International Conferences Interfaces and Human Computer Interaction' in 2023.

We look forward to receiving your revised manuscript.

Kind regards,

Lisa A. Pace

Academic Editor

PLOS ONE

Journal Requirements:

Reviewers' comments:

Reviewer's Responses to Questions

**Comments to the Author**

1. Is the manuscript technically sound, and do the data support the conclusions?

Reviewer #1: Partly

2. Has the statistical analysis been performed appropriately and rigorously? 

Reviewer #1: I Don't Know

3. Have the authors made all data underlying the findings in their manuscript fully available?

Reviewer #1: No

4. Is the manuscript presented in an intelligible fashion and written in standard English?

Reviewer #1: Yes

5. Review Comments to the Author

Reviewer #1: The paper is about user participation in housing design and the development and the use of digital tools and methods - GUI, co-design - to facilitate this. It's a prescient topic and more research in this area is certainly needed. Some development of parts of the paper are required:

Literature review. e.g. from John Frazer and Walter Segal’s digital tools for self-builders in the 1980s to the generative design tools available today e.g. HomeMaker, Bruno Postle (+A Pattern Language, C. Alexander), for clients and other stakeholders e.g. TestFit, Archistar Generative Design. Some discussion of ‘community architecture’, 1980s-90s, ‘planning for real’ events, Habraken, Erskine (Byker Wall), John Turner, 'community architecture' as a continuing movement etc.

The paper is mostly about interaction via GUI with inhabitants and end-users, not about the development of new digital tools. This could be made clearer in the abstract, or introduction. Title ‘… a digital tool’? The use of Figma as an GUI intermediary is novel, relevant and provides some useful insights. It would be helpful to review similar approaches in the literature review.

Figures & Tables. Figures cross-reference check is needed e.g. Fig. 1 lines 329 and 425, Fig. 3. lines 361 and 462. There would appear to be10 images total, however there might be more e.g. Fig. 4 line 513, couldn’t find this. Quite confusing. Table 4, line 607, needs adjusting to clearly show totals and the columns they relate to.

Findings. In general, the evidence which backs up evaluation of the research questions needs to be more thoroughly and carefully presented e.g. it would be helpful if there were more clearly presented graphic/visual data which confirms qualitative outcomes - e.g. charts, graphs – would assist understanding and strengthen the case for evaluations and conclusions drawn.

Providing additional information for co-design users is essential e.g. cost of construction and materials. Consider other information that would assist users’ decision-making e.g. energy use/costs, prefabrication v. site construction, transportation (materials, components, people) costs etc. See how pioneers of this approach, Frazer/Segal, explored and implemented this.

Discussion. Curious to know how regulatory requirements were met by the choices made by users. Also, curious to know how the GUI interfaces with common CAD authoring tools. It would be helpful to classify or group themes that emerged.

Conclusions. This states confirmation of hypotheses – however, the evidence, backed up by a literature review, needs to be presented more fully before these hypotheses can be asserted as confirmed.

6. PLOS authors have the option to publish the peer review history of their article (what does this mean?). If published, this will include your full peer review and any attached files.

Reviewer #1: **Yes: **Robert Doe

---

## [Author Response · Author response to Decision Letter 0]

7 Oct 2024

Dear team,

We took full consideration on the reviewers comments, and as such, we revised the manuscript to adress each point raised by the reviewers and academic editor. The responses are described in both "Response to reviewers" files. We have divided into two files, to separate the responses for the reviewer from the responses for the academic editor, since the latter has information that can disclose the authors identity (the link for the repository hosting the Supporting Information files).

The Supporting information dataset was updated, by including files that were requested on the reviewers comments. Thus, the link for the latest version was updated as well. We have also included the link in the data availability statement at the submission platform.

As requested, we have submitted the Revised manuscript with tracked changes, and an unmarked version of the manuscript. We also reuploaded the images, since we added images in response to the reviewers, and their numbering changed.

The Ethics statement is included in the methods section, as it was previously requested and submitted. The files corresponding to the ethics approval were also previoulsy submitted and mantained in this revised submission.

Thank you very much for the comments, we believe the manuscript evolved for a much better version now.

Kind regards,

Micaela Raposo

---

## [Editor Report · Decision Letter 1]

14 Oct 2024

PONE-D-24-25657R1Customised housing design: Bridging gaps through the interaction with a graphical user interface for users’ collaboration.PLOS ONE

Dear Dr. Raposo,

Thank you for submitting your manuscript to PLOS ONE. After careful consideration, we feel that it has merit but does not fully meet PLOS ONE’s publication criteria as it currently stands. Therefore, we invite you to submit a revised version of the manuscript that addresses the points raised during the review process.

The revised manuscript is considerably improved, and the authors have taken appropriate steps to integrate the suggestions made by the reviewer and the editor. I have some further suggestions, mainly in the title, as minor amendments that need to be implemented.

I am suggesting that you amend the title to improve its readability to: “Bridging the gap in customised housing design: integrating a graphic user interface for user collaboration”.

The manuscript would benefit from thorough proofreading to correct several typos. Some are outlined below (but I am sure there are others that I missed):

p.7 Line 160 “for the [delete their] lack of involvement of the inhabitants……”p.14 Ln326 “Portuguese” [capital P].p14, Ln329: “…through a request [delete requested]………p14 Ln3030: “the main criterion used was that the inhabitants should be the house’s [insert first ever] tenants”p. 42 Ln 40: “………..(e.g. ModRule [44] and HOPLA-Home Planner [29]) involved testing their framework [add with different types of users]”. This sentence seems to be cropped short.p.46, Ln 1027: “….co-creating [amend to:] design for houses in an informed way” At this stage, you are to include in-text references to your own work and insert the Acknowledgement. 

We look forward to receiving your revised manuscript.

Kind regards,

Lisa A. Pace

Academic Editor

PLOS ONE
---

## [Author Response · Author response to Decision Letter 1]

18 Oct 2024

Dear Editor of PLOS ONE,

We are pleased to submit the revised version of the Manuscript, addressing the minor revisions required. 

As suggested, we amended the title and corrected all the points raised in the decision letter. Additionally, we corrected a sentence in the methodology (p.14, Line 308) by substituting “steps 8 and 9” for “steps 9 and 10”, as we detected it was not correct as it was. Some punctuation issues we identifyed were also corrected.

We also have disclosed all the information previously omitted for blind review:

(1) We added the authors' names and affiliations on the title page

(2) We disclosed the name of the institution that provided the Ethics approvement

(3) We added the in-text references to our own work, which were previously omitted throughout the text

(4) We acknowledged that this work is based on the work published and presented at the IHCI 2023 International Conference, on the acknowledgement section

(5) We provided in the Supporting Information section the link for the repository (Zenodo) that hosts the dataset 

Sincerely,

Micaela Raposo, Sara Eloy and Miguel Sales Dias

---

## [Editor Report · Decision Letter 2]

22 Oct 2024

Bridging the gap in customised housing design: integrating a graphic user interface for user collaboration

PONE-D-24-25657R2

Dear Ms. Raposo,

We’re pleased to inform you that your manuscript has been judged scientifically suitable for publication and will be formally accepted for publication once it meets all outstanding technical requirements.

Kind regards,

Lisa A. Pace

Academic Editor

PLOS ONE
---

## [Editor Report · Acceptance letter]

10 Dec 2024

PONE-D-24-25657R2 

PLOS ONE

Dear Dr. Raposo, 

I'm pleased to inform you that your manuscript has been deemed suitable for publication in PLOS ONE. Congratulations! Your manuscript is now being handed over to our production team.

Kind regards, 

on behalf of

Dr. Lisa A. Pace 

Academic Editor

PLOS ONE